# Key transcriptional effectors of the pancreatic acinar phenotype and oncogenic transformation

Ana Azevedo-Pouly[1☯¤a], Michael A. Hale[1☯], Galvin H. Swift[1☯], Chinh Q. Hoang[1¤b], Tye G. Deering[1¤c], Jumin Xue[1], Thomas M. Wilkie[2], L. Charles Murtaugh[3], Raymond J. MacDonald[1]*

**1** Department of Molecular Biology and the Hamon Center for Regenerative Science and Medicine, University of Texas Southwestern Medical Center, Dallas, Texas, United States of America, **2** Department of Pharmacology, University of Texas Southwestern Medical Center, Dallas, Texas, United States of America, **3** Department of Human Genetics, University of Utah, Salt Lake City, Utah, United States of America

☯ These authors contributed equally to this work.
¤a Current address: Department of Surgery, University of Arkansas for Medical Sciences, Little Rock, Arkansas, United States of America
¤b Current address: Vinmec Hi-Tech Center, Vinmec Healthcare System, Hai Ba Trung, Hanoi, Vietnam
¤c Current address: Pfizer, Sanford, North Carolina, United States of America
* amphioxus@me.com

**Data Availability Statement:** All genomic analysis data files are available from the NCBI Gene Expression Omnibus GEO (GEO accession numbers GSE86262, GSE86289, GSE100881).

## Abstract

Proper maintenance of mature cellular phenotypes is essential for stable physiology, suppression of disease states, and resistance to oncogenic transformation. We describe the transcriptional regulatory roles of four key DNA-binding transcription factors (Ptf1a, Nr5a2, Foxa2 and Gata4) that sit at the top of a regulatory hierarchy controlling all aspects of a highly differentiated cell-type–the mature pancreatic acinar cell (PAC). Selective inactivation of *Ptf1a*, *Nr5a2*, *Foxa2* and *Gata4* individually in mouse adult PACs rapidly altered the transcriptome and differentiation status of PACs. The changes most emphatically included transcription of the genes for the secretory digestive enzymes (which conscript more than 90% of acinar cell protein synthesis), a potent anabolic metabolism that provides the energy and materials for protein synthesis, suppressed and properly balanced cellular replication, and susceptibility to transformation by oncogenic Kras$^{G12D}$. The simultaneous inactivation of *Foxa2* and *Gata4* caused a greater-than-additive disruption of gene expression and uncovered their collaboration to maintain *Ptf1a* expression and control PAC replication. A measure of PAC dedifferentiation ranked the effects of the conditional knockouts as Foxa2 +Gata4 > Ptf1a > Nr5a2 > Foxa2 > Gata4. Whereas the loss of Ptf1a or Nr5a2 greatly accelerated Kras-mediated transformation of mature acinar cells in vivo, the absence of Foxa2, Gata4, or Foxa2+Gata4 together blocked transformation completely, despite extensive dedifferentiation. A lack of correlation between PAC dedifferentiation and sensitivity to oncogenic *Kras$^{G12D}$* negates the simple proposition that the level of differentiation determines acinar cell resistance to transformation.

**Funding:** This study was supported by grant R01 DK61220, National Institute of Diabetes and Digestive Diseases, awarded to RJM. The funders had no role in study design, data collection and analysis, decision to publish, or preparation of the manuscript.

**Competing interests:** The authors have declared that no competing interests exist.

## Introduction

Cell-specific lineages leading to highly differentiated cells form during prenatal development through the induction of particular sets of lineage-restricted transcriptional regulators informed by spatially restricted developmental intracellular signaling [1] and the establishment of chromatin architecture distinct for each mature cell-type [2]. Many of the lineage-restricted differentiation transcription factors (dTFs) are retained in the specified adult cells and help maintain the chromatin structure that facilitates transcription of lineage specific genes, stifles transcription of genes specific to other lineages, and alters transcription of some commonly expressed genes for cell-type specific functions. This maintenance of cellular identity and specialized functions is crucial for resistance to cellular stress and injury, as well as diseases such as inflammation and carcinogenesis [3].

Toward understanding the stability of differentiation states in mature organs, we examined the roles that four pancreas-specific dTFs (Ptf1a, Nr5a2, Foxa2 and Gata4) play in the maintenance of the specialized pancreatic acinar cell (PAC) phenotype. These dTFs have crucial developmental roles directing cell fate decisions, expansion of pancreatic precursor populations, and the final stage of acinar cell differentiation [4]. Their presence is maintained in mature PACs, but much less is known of their roles to preserve the differentiation status and homeostasis of the adult cells.

Ptf1a is a bHLH protein that helps establish and maintain the commitment to pancreatic fate at the onset of pancreatogenesis [5–7] and later directs the resolution of the acinar cell lineage from the tripotent progenitor cell population for acinar, islet and ductal cells [8–10]. Human mutations that prevent *Ptf1a* expression in the early pancreatic endoderm cause pancreatic agenesis and, consequently, neonatal diabetes [11]. The transcriptional activity of Ptf1a requires its inclusion in a unique, stable complex with two other DNA-binding proteins: a common bHLH protein such as E47 and either Rbpj or Rbpjl [12,13]. Rbpj is the transcriptional effector of canonical Notch signaling [14] present in the trimeric PTF1-J complex for cell-fate instruction during the early stages of pancreatic development [15]. *Rbpjl* encodes a pancreas-restricted paralog of Rbpj [16] that replaces Rbpj in the PTF1-L complex [12] and drives the final stage of acinar cell differentiation [17]. In adult acinar cells, Ptf1a is required for maintenance of proteostasis, the high state of differentiation, cellular identity, and resistance to Kras-mediated oncogenesis [18–20].

Nr5a2 (Lrh1), a member of the family of DNA-binding nuclear hormone receptors [21], specifies select domains of embryonic development [22,23], organogenesis [24], and aspects of metabolic regulation in adult vertebrates [25–27]. During pancreatic development, Nr5a2 is required for the expansion of the tripotent progenitor cell population and the proper allocation of precursors to acinar fate, as well as the complete differentiation of acinar and ductal cells and the growth and maintenance of acinar tissue mass and homeostasis [24,28,29]. In adult PACs, Nr5a2 cooperates with PTF1-L to control expression of the digestive enzyme genes [30]. Nr5a2 has been implicated in pancreatic and intestinal cancers [31–34]. It increases growth of pancreatic cancer cells in culture [35] and bestows resistance to oncogenesis by countering acinar cell inflammation [28].

Foxa2 and its paralogue Foxa1 are important to early pancreatic development and the proper formation of all three pancreatic epithelial cell-types [36]. Foxa2 is needed selectively for the subsequent formation of alpha-cells [37] and maintenance of beta-cell function of the endocrine compartment [38], but its role in acinar and ductal cell formation and maintenance of mature cells is not yet established. Observations that Foxa2 may enhance stemness and tumorigenicity [39] and suppress epithelial-to-mesenchymal transition in pancreatic ductal

adenocarcinoma (PDAC) cells [40] suggest that Foxa2 may play different roles at multiple stages of tumorigenesis [41].

Gata4 and Gata6 are important and partly redundant for early pancreatic development [42–44] and subsequently required for proper exocrine tissue formation [45,46]. The inactivation of *Gata6* during early fetal pancreatic development disrupts acinar differentiation and sensitizes affected cells to transformation by oncogenic Kras [47]. In this report we describe how inactivating *Gata4* selectively in the adult acinar pancreas undermines acinar differentiation and protects from Kras-mediated transformation.

The Foxa and Gata4/6 factors are developmental pioneer transcription factors [48,49] that help establish chromatin architecture in the early endoderm, restrain developmental potential to gut fates, and provide a foundation for regionalization of the gut endoderm, including the pancreas [50,51]. Here we show that Foxa2 and Gata4 cooperate to sustain the acinar differentiated state by maintaining *Ptf1a* expression and limiting acinar replication.

The disruption of acinar cell differentiation and identity due to stress [52,53], inflammation [28], injury [54,55], or loss of key regulators of differentiation [56,57] confers susceptibility to oncogenic transformation [20,58,59]. For example, serial pharmacologic doses of the acinar secretagogue cholecystokinin or its analog caerulein induce inflammation, degrade differentiation, and weaken cell-identity through a process termed acinar to ductal metaplasia [60]. Undergoing this process sensitizes acinar cells to transformation by *Kras* bearing oncogenic mutations [61,62]. Similarly, the engineered inactivation of *Ptf1a* disrupts the differentiation status of adult acinar cells directly and greatly sensitizes them to transformation by oncogenic Kras$^{G12D}$ [20]. Re-activation of *Ptf1a* expression can restore differentiation and reverse the precancerous transformation [63]. The influence on oncogenic transformation of conditional inactivation of the other dTFs in mature acinar cells has not been previously reported.

We show that although all four dTFs maintain the differentiated state of PACs, they have qualitatively and quantitatively different roles. Furthermore, only Ptf1a and Nr5a2 suppress oncogenic transformation in vivo, and contrary to expectations, Foxa2 and Gata4 facilitate oncogenic transformation in vivo. These findings refute the hypothesis that the level of differentiation alone determines acinar cell resistance to transformation.

## Results

### Identification of Ptf1a, Nr5a2, Foxa2 and Gata4 as dTFs for the mature PAC

We applied four criteria to distinguish sequence-specific DNA-binding transcription factors that likely define the differentiated state of the mature pancreatic acinar cell. First, restriction to acinar cells and few (or no) other differentiated cell-types; restricted expression is key to cell-type specific control [64]. Second, presence at a higher level than most other TFs; a high nuclear concentration saturates cell-specific gene transcriptional control regions to ensure effective control of differentiation properties [65]. Third, participation in specifying pancreatic acinar fate or controlling acinar differentiation during prenatal development. And fourth, presence in chromatin at most PAC-specific transcriptional enhancers. Indeed, an initial analysis of Ptf1a binding sites detected nearby Foxa (*p* value = e-353) and Gata4/6 (*p* = e-182) consensus binding sequences at high frequencies in PAC-specific transcriptional enhancers [66]. Applying these criteria, we identified four most promising acinar candidate dTFs—Ptf1a, Nr5a2, Foxa2 and Gata4 (**Table 1** and S1 Fig).

**Broad regulatory functions of Ptf1a, Nr5a2, Foxa2 and Gata4.** To establish the regulatory responsibilities of the four dTFs, we monitored the changes in the mRNA population after inactivation of each dTF gene selectively in the acinar cells of the adult mouse pancreas

**Table 1. DNA-binding transcription factors with high and restricted expression for pancreas, parotid gland, and liver.**

| Pancreas | | | | Parotid | | | | Liver | | | |
|---|---|---|---|---|---|---|---|---|---|---|---|
| rank | TF | rpkm | #/35 [a] | rank | TF | rpkm | #/35 | rank | TF | rpkm | #/35 |
| 2 | Rbpjl | 57 | 11 | 2 | Mist1 | 171 | 2 | 3 | Hnf4a | 176 | 7 |
| 4 | Mist1/Bhlha15 | 51 | 7 | 3 | Pax9 | 156 | 6 | 6 | Cebpb | 109 | 8 |
| 8 | Ptf1a | 15 | 2 | 4 | Etv1 | 111 | 15 | 12 | Nr1i3 | 63 | 2 |
| 15 | Gata4 | 7.7 | 10 | 8 | Sox10 | 73 | 9 | 13 | Nr1h4 | 62 | 6 |
| 26 | Foxa2 | 5.7 | 5 | 16 | Cebpb | 46 | 8 | 23 | Nr0b2 | 40 | 4 |
| 31 | Nr5a2 | 5.2 | 7 | 18 | Barx2 | 41 | 3 | 25 | Hhex | 37 | 9 |
| 37 | Foxa3 | 4.8 | 3 | 25 | Pitx1 | 33 | 7 | 26 | Nr1i2 | 33 | 3 |
| | | | | | | | | 31 | Foxa3 | 29 | 3 |

Complete lists of the highest 70 TF mRNAs are in S1 Fig. Parotid RNAseq results by GH Swift, RJ MacDonald and T Wilson; GSE102788. Liver RNAseq results from Mortazavi et al. [67].

[a] #/35, number of organs/tissues with one or more Expressed Sequence Tags in NCBI UniGene EST Profiles (https://www.ncbi.nlm.nih.gov/unigene/) out of the 35 organs/tissues with >6,000 total ESTs

using floxed dTF genes and *Ptf1a^CreERT* [68] with tamoxifen-induction (**Fig 1A**) (see Methods for detailed genotypes). The conditional knockout of each floxed dTF locus (designated Foxa2-cKO, etc.) was highly efficient—91.8 to 99.8% (S1 Table).

The cKOs of the four dTFs had distinguishing quantitative and qualitative effects on the differentiated functions of acinar cells. The relative regulatory importance of the four dTFs was inferred from the number of genes affected by the individual inactivation of their genes: *Ptf1a*, 3,334; *Foxa2*, 2,096; *Gata4*, 1,428; *Nr5a2*, 1,241 (**Fig 1B**; all results in S3 Table). The total non-redundant gene set affected by at least one of the four dTFs was 5,473 genes, nearly half of the total pancreas-expressed genes detected by RNA-seq. Generally, the numbers of genes with decreased mRNA were about equal to those with increased mRNA (**Fig 1C**). *Ptf1a* participated in the most pairwise co-regulation, sharing 873 affected genes with *Foxa2*, 674 with *Gata4* and 588 with *Nr5a2* (**Fig 1D**). Slightly more than half of the *Ptf1a*-affected genes were not affected by any of the other three dTFs. In comparison, the other dTFs have singular responsibility for only 27–42% of their affected gene sets. Thus, *Ptf1a* has the most regulatory breadth alone and in collaboration with the other dTFs. Each altered dTF mRNA population could be readily distinguished from the others by multidimensional scalar analysis (**Fig 1E**). The results from further analyses (below) indicate that a large component of the horizontal displacement in the multidimensional scaling plot correlates with acinar cell differentiation and the vertical includes cell replication.

The dTFs share partly overlapping control of PAC differentiation, replication, and metabolism (**Fig 1F and 1G**). All contribute to the acinar specific transcription of the 35 secretory enzyme genes as well as 23 additional acinar restricted genes for post-translational processing, intracellular transport, and exocytosis. Ptf1a dominates control of the machinery and provisions for secretory protein synthesis. Ptf1a, Nr5a2 and Foxa2 have broad effects on genes of the highly anabolic metabolism of acinar cells. Foxa2 plays an important role in regulating acinar cell replication. Ptf1a and Foxa2 have the most influence on the expression of the complement of DNA-binding transcription factors (see also S2 Fig). Gata4 preferentially affects the system for processing the secretory enzymes in the rough endoplasmic reticulum and appears to restrain more genes associated with acinar differentiation than the other dTFs. The compositions of signature sets of genes [70] that distinguish the cKO transcriptomes from one another confirmed the emphasis of *Ptf1a* control of genes for protein production as well as

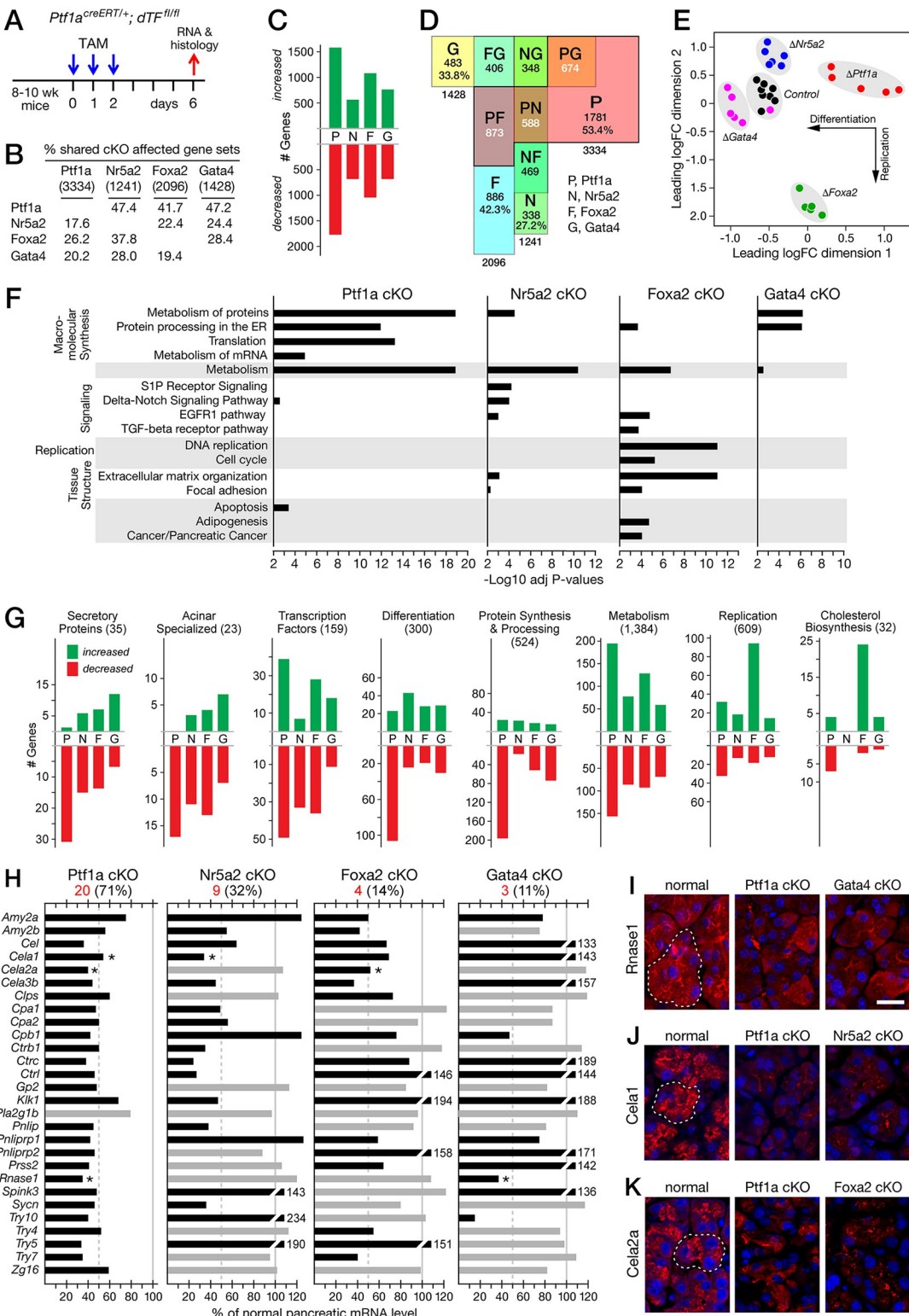

**Fig 1. Effects of dTF gene inactivation on the predominant cellular processes of pancreatic acinar cells.** **A**. Treatment scheme for tamoxifen (TAM) induced CreERT inactivation of floxed dTF genes. **B**. Numbers of genes with affected expression for each dTF-cKO (parentheses) and pairwise percent of co-affected genes. **C**. Comparisons of the number of genes for each dTF-cKO with increased (*green bars*) and decreased (*red bars*) mRNA levels (cKOs: P, Ptf1a, N, Nr5a2, F, Foxa2; G, Gata4) **D**. Semiquantitative Venn diagram of the overlapping sets of the dTF-affected genes. **E**. Multidimensional scaling analysis for the

four mutant pancreatic transcriptomes and control. **F**. Acinar processes enriched for genes affected by the dTF-cKOs from analysis using the ConsensusPathDB [69]. **G**. Numbers of affected genes encoding acinar specific proteins and proteins of specialized cellular processes. Lists of the genes for the processes were assembled from a combination of Reactome, KEGG and Wiki Pathways (S2 Table). **H**. Levels of the mRNAs for the 28 highest expressed acinar secretory proteins relative to levels in control mice (*Ptf1a*$^{CreERT/+}$ with TAM-treatment). *red*, number of mRNAs (and %) at half or less of normal levels. *Grey bars*, DE >0.01 fdr; *black bars*, <0.01 fdr. **I**-**K**. Immunofluorescent detection of the specialized differentiation markers Rnase1, Cela1 and Cela2a. *Dashed lines* outline examples of normal acini. Bar = 20 μ for I-K.

*Foxa2* suppression of replication genes and uncovered stimulatory effects of *Nr5a2* on genes of replication (S3 Fig).

The roles each dTF plays in defining the differentiated state of PACs–acinar specific gene expression, prodigious protein synthesis, regulated cell replication, specialized anabolic metabolism, and resistance to oncogenic transformation—are described in turn.

**Each dTF is required to maintain PAC differentiation.** Changes in the acinar differentiated state can be estimated from effects on the production of the secretory enzymes and the other PAC-restricted gene products. All four dTF-cKOs severely affected the level of the mRNAs for the 35 acinar secretory proteins, in the order Ptf1a > Nr5a2 > Foxa2 > Gata4 (**Fig 1G**). The magnitude of the effects on individual secretory protein genes were similarly graded (**Fig 1H**). The mRNAs for 20 of the most highly produced secretory enzymes diminished by 50% or more by the depletion of Ptf1a. When examined, the levels of immunostaining protein reflected the decreased mRNA levels (**Fig 1I–1K** for ribonuclease, elastase 1 and elastase 2a). Although all four dTF-cKOs affected the state of acinar differentiation, only the Ptf1a-cKO activated nonpancreatic genes. Of the fourteen stomach- or intestine-restricted genes derepressed by the loss of Ptf1a [18], only two were affected by the loss of Foxa2, one by Nr5a2 and none by Gata4 –and for those, the induction levels were one or two magnitudes less than for Ptf1a (S4 Table).

**Extensive functional collaboration among the dTFs.** To estimate the fraction of regulatory effects that are due to direct transcriptional control, we examined the genome-wide binding of the four dTFs to isolated pancreatic chromatin and their association with specific target genes (**Fig 2**). To focus on functional dTF binding, we restricted further analysis to sites in active regulatory domains (ARDs), defined as extended regions of histone H3K4-dimethylation (H3K4me2) that also have bound RNA polymerase II (RNAPII). H3K4me2 marks active enhancers and promoters [71], and RNAPII is an independent marker of eRNA synthesis at enhancers [72] as well as promoter regions near the transcriptional start sites of active genes. Only dTF-bound sites in ARDs were considered in further bioinformatic analyses (**Fig 2A**).

To establish the importance of dTF binding in pancreatic ARDs, we first verified the presence of Ptf1a, Nr5a2, Foxa2 and Gata4 bound to ARDs of individual acinar secretory enzyme genes and other key genes associated with the acinar phenotype (S4 Fig) that are highly dependent on one or more dTF (**Fig 1H**). More broadly, high fractions (43.8 to 64.8%) of all genes affected by the inactivation of a dTF gene had associated ARDs with the cognate dTF bound (**Fig 2B** and S5 Table). The four dTFs frequently colocalized to ARDs (**Fig 2D and 2E**), characteristically associated with the presence of RNAPII binding and positioned within the interval between H3K4me2-marked nucleosomes. For example, 65% of the Foxa2-bound sites were within 200-bp of bound Gata4 and of those more than half had all four dTFs bound (**Fig 2E**). Examination of the DNA sequences surrounding the binding sites of each dTF confirmed the common presence of consensus sequences for the other three dTFs (**Fig 2C**), indicating direct, co-localized binding to DNA in the functional design of acinar enhancers. All but Foxa2-- bound sites have the Rbpjl binding motif enriched highly. Co-enrichment of the Ptf1a E- box and the Rbpjl motif for the dTF peaks is expected for the presence of the Ptf1a:E2protein:

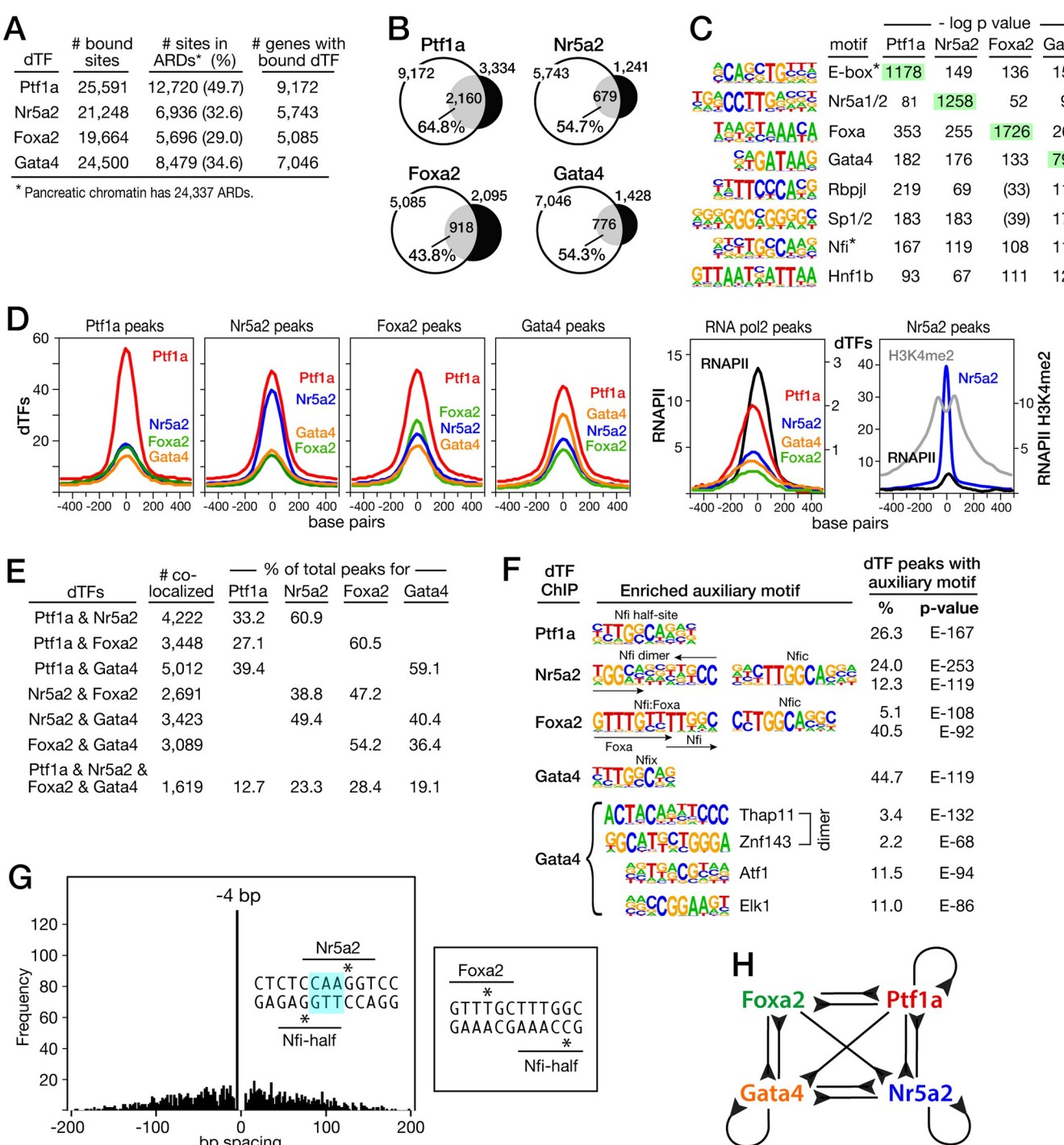

**Fig 2. dTF binding to regulated acinar promoters and enhancers.** **A**. Genome-wide numbers of total binding sites, sites (and % of total) in an Active Regulatory Domain (ARD), and genes associated with ARD peaks using GREAT [73] for the four dTFs. **B**. 'Regulated Genes': percentages of DE genes for each cKO with the cognate dTF bound to an ARD in normal pancreas. **C**. *de novo* enriched oligo-sequences (enrichment p values <1E-50) within centered 200-bp regions of dTF peaks. **D**. Highly coincidental dTF binding in ARDs and association with RNAPII and H3K4-dimethylation. **E**. Pairwise colocalization of dTFs in ARDs; includes those also in three-way (not shown) and four-way combinations. **F**. Auxiliary TF binding site motifs in dTF peaks identified by de novo discovery. **G**. Precise spacings at paired Nfi and Nr5a2 (4-bp) or Foxa2 at binding sites; *asterisks*, centers of consensus binding motifs. **H**. Cross-binding of dTFs to ARDs of each other's genes.

Rbpjl form of the trimeric PTF1 complex. Examples of clustered dTF-binding are displayed in subsequent figures. The high numbers of such clustered dTF-binding sites in transcriptionally active ARDs indicate extensive functional collaboration among the dTFs, which extends to cross-binding among the dTFs with their genes (**Fig 2H**).

Motifs for other families of transcription factors were found at dTF-bound sites. For example, the consensus binding-site sequences of the Nfi family of factors were commonly enriched at the bound sites of each dTF, and consensus binding motifs of Thap11 and Znf143, Atf1 and Elk1 were enriched at Gata4-bound sites (**Fig 2F**). The presence of combined sites for Foxa2 or Nr5a2 in precise sequence register with Nfi sites (**Fig 2G**) suggests cooperative binding by these two dTFs with Nfi proteins. In general, genes with Nfi motifs at dTF-bound sites are enriched for the same cellular pathways as the individual dTFs, although there are exceptions: cancer, apoptosis, intermediary metabolism, and PDGF signaling (**S5 Fig**). As for other cell-types [74–76], it appears that the Nfi proteins can act as selecting or nonselecting cofactors for acinar dTFs depending on the target gene and its regulatory context.

**Differential effects on acinar protein synthesis.** Whereas *Ptf1a* broadly regulates multiple mechanistic layers of PAC secretory protein synthesis, *Nr5a2*, *Foxa2* and *Gata4* have more restricted roles (**Fig 3A and 3C**). To assess the consequences to protein synthesis rates for each cKO, we measured the incorporation of $^{35}$S-methionine and $^{35}$S-cysteine into protein during the incubation of micro-dissected lobules from pancreases of normal and cKO mice (**Fig 3D**). To account for differences in acinar cell size among the dTF-cKOs and normal mice [18], we normalized incorporation to cell-number based on the amount of tissue DNA in each assay. The Ptf1a-, Nr5a2-, Foxa2- and Gata4-cKOs decreased protein synthesis, the most meaningful acinar characteristic, by 50, 48, 13, and 44%, respectively (**Fig 3E**).

The effects on protein synthesis can be attributed to several sources: the depletion of secretory protein mRNAs, altered Myc expression that broadly affects protein synthetic machinery, depletion of mRNAs for specific regulatory or rate-limiting proteins, and altered post-translational modification of protein synthesis regulators.

Because the mRNAs for the acinar secretory proteins compose >90% of the total mRNA population of murine pancreas [77], changes in the transcription of these few genes alone can have exaggerated effects on the total amount of mRNA and consequently would be expected to have proportional effects on the overall rate of protein synthesis. In this regard, the cKOs of *Ptf1a*, *Foxa2* and *Gata4* decreased the size of pancreatic mRNA populations by 33, 39 and 12%, respectively, and the Nr5a2-cKO population increased by 11% (**S6 Fig**) (largely due to an increase of amylase mRNA). These effects did not correlate with the measured rates of protein synthesis for the four dTF-cKOs (**S6B Fig**). Most noteworthy is the 11% increased mRNA population for Nr5a2-cKO compared to the 48% decrease of protein synthesis. Other regulatory mechanisms are likely the primary cause of the idiosyncratic changes in protein synthesis. Indeed, the effects on protein synthesis best correlate with changes of expression and post-translational regulation of genes known to control the rate of translation in PACs [78–80] (see below).

Myc increases cell-growth principally through the induction of genes for protein synthesis and metabolism [81,82]. *Myc* is required for proper acinar development and its expression continues in mature acinar cells [18,83]. The massive size, protein synthesis and anabolic metabolism of acinar cells may be attributed in part through Myc control [18,84]. *Ptf1a* inactivation decreased Myc expression by 77% (**Fig 3F**) and the expression of 44 of the 130 hallmark up-regulated Myc target genes [85] also decreased. In contrast, Nr5a2-, Foxa2- and Gata4-cKOs decreased only 14, 4 and 14 hallmark genes, respectively (**S6C Fig**). Of the 28 protein synthesis genes in this Myc target set, inactivation of *Ptf1a* decreased 16, and of *Nr5a2*

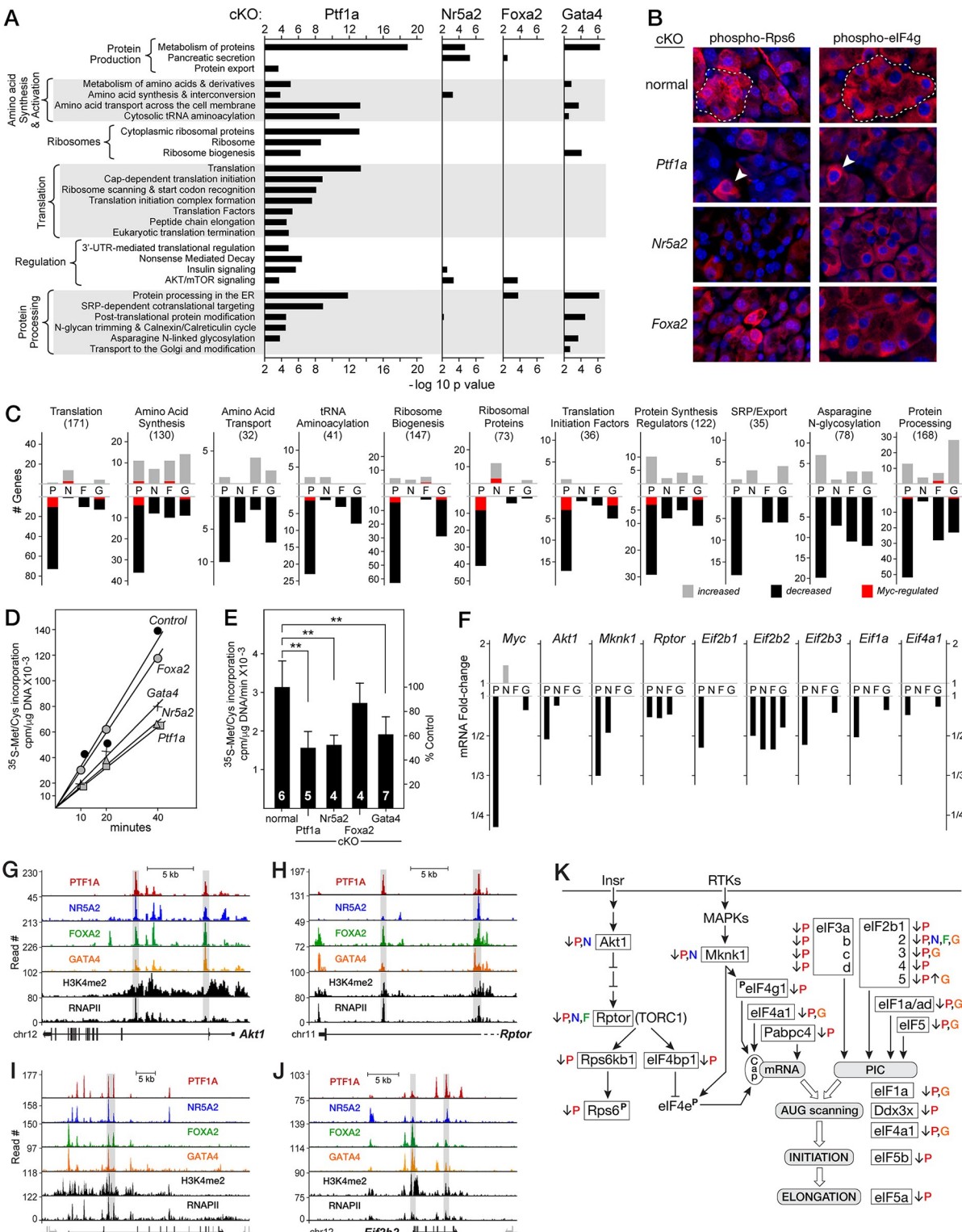

**Fig 3. Effects of dTF gene inactivation on acinar protein synthesis. A**. Gene sets for protein synthesis pathways preferentially affected by dTF gene inactivation. **B**. Immunofluorescence detection of phosphorylated Rps6 and eIF4G in normal and dTF-cKO pancreases. *Dashed outlines*, normal acini; *arrowheads*, cells that escaped *Ptf1a*-deletion. **C**. Numbers of genes affected by the dTF-cKOs that associate with sub-processes of protein production. *Parentheses*: total number of genes in each category (see S2 Table); *red inserts*, number of Myc hallmark target genes. P, Ptf1a; N, Nr5a2; F, Foxa2; G, Gata4 cKOs. **D**. Rates of protein synthesis in short-term cultures of pancreatic lobules measured

by incorporation of $^{35}$S-methionine/cysteine per μg DNA per minute (representative experiments for each cKO). Lines are the linear regression fits. **E**. Summary of all experiments measuring rates of protein synthesis for the dTF-cKOs. Numbers in bars are the number of mice for each genotype. Student's t-test results with SD; ** p value <0.01. **F**. Changes of mRNA levels (RNA-seq) for key regulators of protein synthesis; no bar indicates values with an fdr >0.01. **G-J**. ChIP-seq profiles for the binding of the dTFs to four key target genes for protein synthesis. *Shaded bars*, ARDs. The *grey* gene fragments are the ends of neighboring genes: *Mobkl2c* and *Kncn* for *Mknk1* and *Mlh3* for *Eif2b2*. **K**. Regulatory and potential rate-limiting steps of acinar protein synthesis. Colored letters indicate which cKOs affected mRNA levels.

three, *Foxa2* none, and *Gata4* three. Although some of the effect of *Ptf1a* can be mediated through *Myc*, it is less so for the other dTFs (**Fig 3C**, *red*).

Depletion of mRNAs for specific regulatory proteins and altered post-translational modifications contribute to the effects on protein synthesis. Components of the Akt/Rptor and RTK/MAPK signaling pathways, which control the rate of protein synthesis in PACs [80,83], are diminished in the Ptf1a- and Nr5a2-cKOs (**Fig 3K**). Decreased levels of mRNAs for Akt1, Rptor and Mknk1 (**Fig 3F**) would be expected to lead to lower phosphorylation of their targets that control the rate of protein synthesis, e.g., Rps6, eIF4e, and eIF4g. The loss of Ptf1a affected the mRNA levels of 20 additional regulators of translation; the loss of Nr5a2 affected only four, Gata4 six, and Foxa2 two (**Fig 3K**). The knockouts of *Nr5a2* and *Gata4*, which have similarly diminished protein synthesis rates to the inactivation of *Ptf1a*, share modest decreased expression of *Rptor* and *Eif2b1* or *Eif1b2/3*, *Eif1a* or *Eif4a1* (**Fig 3F**). The loss of Gata4 had additional effects on genes of amino acid transport, tRNA aminoacylation, ribosome biogenesis, and translational regulators (**Fig 3C**), which could restrain protein synthetic rates. These observations indicate that the dTFs optimize acinar protein synthesis through effects on partly overlapping sets of regulators and supplies of synthetic precursors such as the amino acids and their acylated-tRNAs.

The dTFs control most of these target genes directly by binding to the pancreatic ARDs of these genes, including *Akt1*, *Raptor*, and *Mknk1* [83], which control Rps6 phosphorylation and Eif4e activity [78–80], *Eif2b2* (affecting activity of the translational preinitiation complex) (**Fig 3G–3J**), as well as genes for supporting processes of protein synthesis such as aminoacyl activation of tRNAs (e.g., *Wars*) and substrate transport (e.g., the glutamine arginine transporter *Slc7a8*/LAT2) (**S6D and S6E Fig**). The 23 protein synthesis regulators shown in **Fig 3K** were affected by the deletion of at least one dTF, and their genes have the corresponding dTFs bound to an associated ARD. Indeed, most have all four dTFs bound within an ARD.

Effects on the phosphorylated status of translational regulators are most prominent for the Ptf1a- and Nr5a2-cKOs. Whereas the mRNAs for Rps6 and Eif4g are not or only marginally affected, decreased phosphorylation of these two regulators correlates with the decreased expression of their kinases, Akt1/Rptor/Rps6kb1 and Mknk1, respectively (see **Fig 3B and 3F**). Phosphorylation of Rps6 (**Fig 3B and 3K**) is a measure of PIP3/AKT/mTOR signaling, which is high in PACs [86]. Phosphorylation by Mknk1 enhances the activity of Eif4e [83], which stimulates translational initiation in acinar cells via binding the mRNA cap [87]. Phosphorylation of the Eif4g (**Fig 3B and 3K**) subunit of EIF4F [88] is a readout of acinar Mknk1/2 activity [83]. Thus, the dTFs enhance the transcription of genes encoding key protein kinases that support the activating phosphorylation of known regulators of PAC translation.

Fgf21 is an autocrine factor that potentiates protein production and secretion [89] and suppresses ER stress [90,91]. The high and selective production of Fgf21 in pancreatic acinar cells is specified by an ARD that is bound and regulated by Ptf1a, Foxa2 and Gata4 (**S6F Fig**). In summary, the dTFs maintain the high level of protein production by driving the expression of genes encoding protein synthetic machinery, key stimulators of translation, the acinar fitness factor Fgf21, and, as will be discussed below, a strong anabolic metabolism that provides the substrates for protein synthesis.

**Acinar cell replication.**    All four dTFs contribute to the regulation of replication (**Fig 4A**), which maintains acinar cell mass in the adult pancreas [92]. The inactivation of *Nr5a2* decreased PAC replication the most, even though few genes were affected. However, the mRNA levels of three key regulatory factors were altered: Mcm2 and Mcm6 of the hexameric complex involved in the initiation of replication decreased to 72% and 43%, and the cell cycle inhibitor Cdkn1c increased 5.7-fold. The Ptf1a-cKO decreased cycling of the acinar cells that lose Ptf1a [18] and this effect is detectable in the fewer number of cells with the replication factor Mcm2 (**Fig 4E**). *Foxa2*-inactivation strongly affected genes of DNA replication and cell cycle control (**Fig 4B and 4C**). Most of the affected genes had increased mRNA levels (**Fig 4D**), and the fraction of acinar cells cycling correspondingly increased (**Fig 4A**). Expression of genes for key effectors and regulators of replication Mcm2 (DNA synthesis), Cdt1 (initiation of replication), and Jun increased in the absence of Foxa2 (**Fig 4D**). The higher mRNA level for Mcm2 was reflected in greater numbers of acinar cells with detectable Mcm2 protein (**Fig 4E**).

Examination of the effects of pairwise inactivation of dTF genes (discussed below) revealed forty replication and cell cycle genes whose induction by *Foxa2*-inactivation depended on the continued presence of Gata4 (**Fig 4D**). Only four of these genes were affected by the loss of the other dTFs alone. Nearly all had Foxa2 and Gata4, but not Ptf1a or Nr5a2, co-bound to a linked ARD (**Fig 4D** and **4H-4J**). In addition, the pattern of chromatin modification and the distribution of bound RNAP2 and of dTFs were unusual compared to most dTF-regulated genes: binding was restricted to the region surrounding the transcriptional start site; no remote enhancer-like regions were evident. In contrast, the genes of other common regulators of replication (*e.g.*, Tfdp1, Jun and Plk2) were bound and regulated by the other dTFs as well (**Fig 4D**) and by distant ARDs (e.g., **Fig 4F and 4G**). The principal regulatory scheme for this subset of replication genes may be surmised: activation by promoter-bound Gata4, which is normally restrained by Foxa2 and largely independent of Ptf1a and Nr5a2 (**Fig 4K**). This suggests a mechanism for controlling acinar replication by altering the activity of either Foxa2 or Gata4.

**The dTFs control many genes of acinar metabolism.**    The extensive effects of Ptf1a, Nr5a2 and Foxa2 on genes of metabolism (**Fig 5A and 5B**) emphasize the central importance of a powerful metabolism to acinar cell function. 78 of the 156 metabolic genes that decreased expression in the Ptf1a-cKO are genes that increased during the final stage of acinar differentiation from embryonic day 18.5 to adult. The dTFs have inordinate effects on genes encoding enzymes of intermediary metabolism (**Fig 5C**) through binding ARDs associated with those genes (e.g., **Fig 5D–5G**). The dTFs enhance expression of genes for glutamine uptake and catabolism, which feeds the TCA cycle [93], a hyperactive methionine cycle [94], the biosynthesis of arginine and proline, the metabolism of branched chain amino acids, and creatine biosynthesis (**Fig 5C**). Moreover, 13 of the 16 most highly expressed nutrient transporter genes were affected by dTF knockouts. Each has at least three of the four dTFs bound at one or more gene-associated ARDs; *e.g.*, transporters of important acinar metabolites (glutamine transporters Slc1a5, Slc38a3, Slc38a5 and hexose Slc50a1), protein synthesis substrates (e.g., amino acid transporters Slc7a8, Slc3a2, Slc6a9, Slc7a5, Slc38a10), and divalent cations critical to ribosomal function ($Mg^{+2}$ Slc41a1 and $Zn^{+2}$ Slc39a5). Broad control by dTFs enhances expression of metabolic genes for substrates and cofactors of protein synthesis and energy for exocytosis.

The pancreas is a major site for production of the metabolite guanidinoacetate [95], the precursor to creatine. And the phosphorylated form of creatine is an energy source for acinar exocytosis [96,97]. The enzyme for guanidinoacetate biosynthesis, Gatm, is maintained at high levels by Ptf1a, Foxa2 and Gata4 (**Fig 5C**) bound to transcriptional regulatory regions (**Fig 5F**). The other enzyme for creatine biosynthesis, Gamt, methylates guanidinoacetate to form

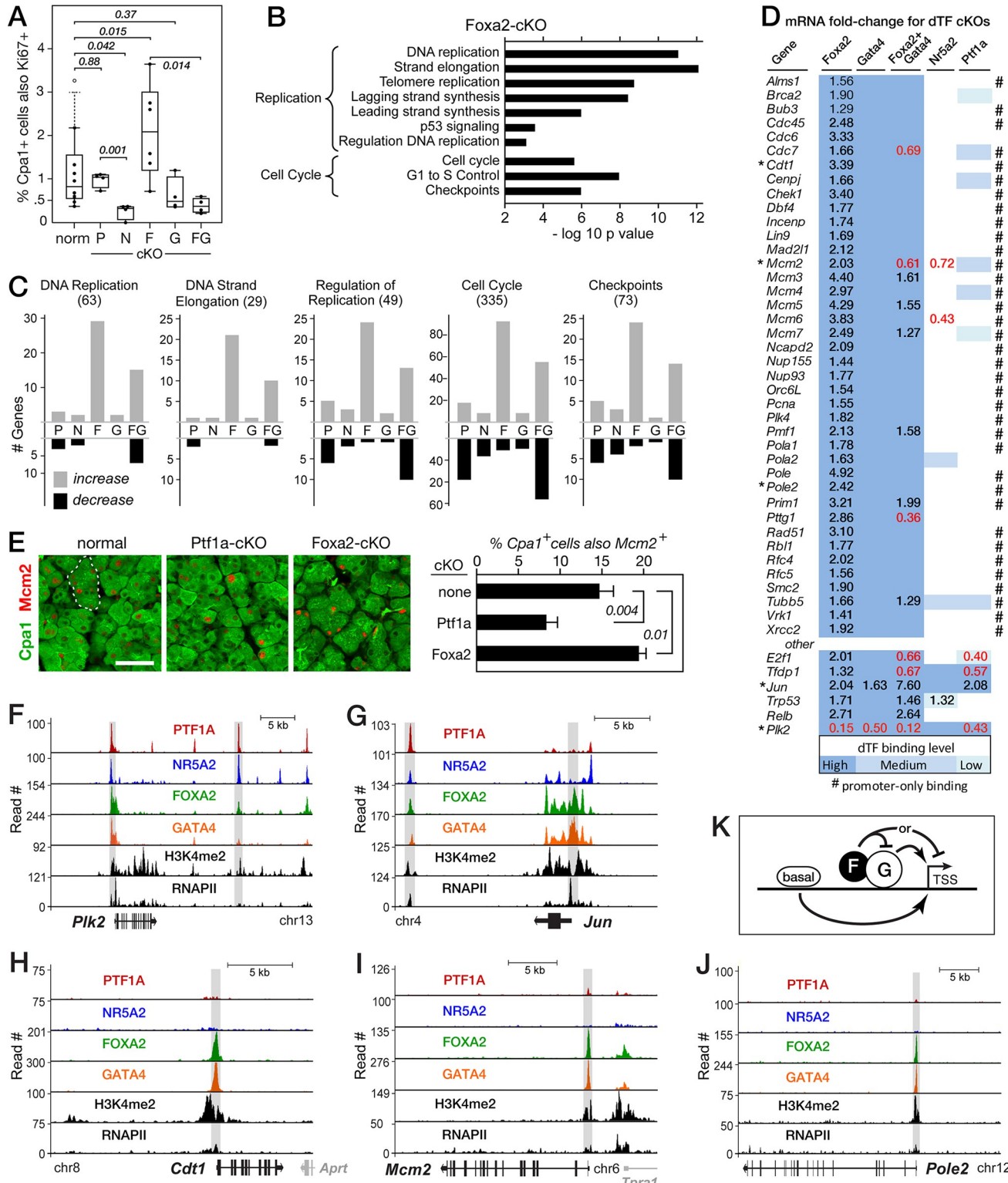

**Fig 4. *Foxa2* suppresses acinar DNA replication and the cell cycle.** **A**. Effects of dTF gene inactivations on the proportion of Cpa1+ acinar cells with Ki67 immunostaining. P-values from pairwise t tests (± S.D.). **B**. Replication pathways enriched for genes misregulated upon *Foxa2* inactivation. **C**. Numbers of genes associated with DNA replication and the cell cycle that were affected by the cKOs. *Parentheses*, total numbers of genes in each category (S2 Table). **D**. Fold-changes of the mRNAs for genes induced upon *Foxa2* inactivation that regulate DNA replication or the cell cycle; fdr values <0.01. #, regulated genes with dTFs bound selectively at transcriptional start sites; *asterisks*, genes and proteins further analyzed in panels E-J. **E**. Immuno-

fluorescence to quantify the fraction of acinar cells with Mcm2. Size bar, 50 μ; *dashed outline*, normal acinus; t-test p values. **F, G**. Examples of replication genes bound and regulated by multiple dTFs. **H-J**. Genes with interactive Foxa2 and Gata4 regulation alone and promoter-restricted binding. *Shaded bars*, ARDs. **K**. Simple model for Foxa2 and Gata4 collaboration at target promoters (see text).

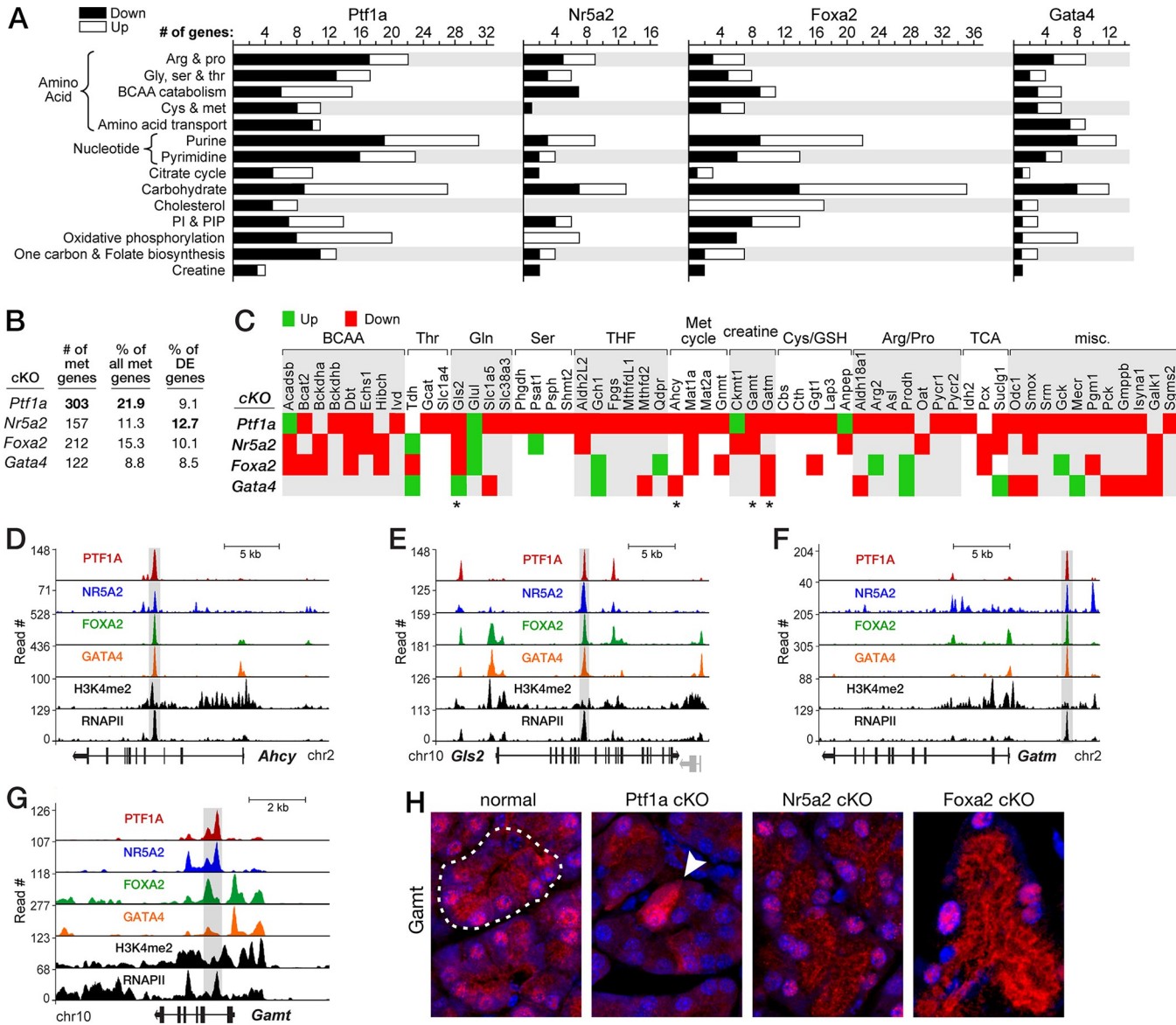

**Fig 5. Specialized PAC metabolism is maintained by the dTFs. A**. The metabolic focus of acinar cells as judged by the enrichment of genes for particular metabolic functions in dTF-cKO pancreases and based on ConsensusPathDB analyses [69]. *Filled bars*, numbers of genes with decreased mRNA; *open bars*, numbers with increased mRNA. **B**. Quantification of the relative importance of metabolic gene control by the four dTFs. Total number of affected metabolism genes based on 1386 genes of KEGG metabolism, the percentage of affected genes, and the percent of total affected genes for each cKO that are classified for metabolic function. **C**. Differentially expressed genes of intermediary metabolism, with emphasis on genes affected by multiple dTF-cKOs. *Asterisks*, genes and proteins reported in panels D-H. **D-G**. Multiple dTF binding at active enhancers of key metabolic genes *Ahcy*, *Gls2*, *Gatm* and *Gamt*. The *grey* gene in panel E is *Sprdy4*. *Shaded bars*, ARDs. **H**. Immunodetection of cytosolic and nuclear Gamt (*red*); *blue*, nuclear DAPI. *Dashed outline*, acinus; *arrowhead*, *Ptf1a*-deletion escapee with nuclear and cytoplasmic Gamt.

creatine; the ARD associated with its gene is bound (**Fig 5G**) and up-regulated by Ptf1a and Nr5a2 (**Fig 5C**). The amount of Gamt in acinar cells of normal, Ptf1a-, and Nr5a2-cKO mice mirrored the level of its mRNA (**Fig 5H**). Gamt is present in both nuclear and cytoplasmic compartments, suggesting that Gamt may have a nuclear substrate as well. The level of Gamt in Ptf1a-cKO pancreas decreased similarly in both compartments.

To summarize the transcriptomic results: the four dTFs control a very large fraction of the acinar transcriptome by activating the expression of acinar specific genes and by selectively altering the expression of many commonly expressed genes to empower specific cellular processes greatly enhanced in acinar cells. Indeed, the specialized acinar cell functions demand high metabolic activity. To delve into the nature of their collaborative regulatory strategies, we next analyzed the genetic interactions among the dTF genes and examined the role their binding plays at the transcriptional enhancer of a key target gene (*Bhlha15/Mist1*) as an illustrative example.

**Regulatory relations among the dTFs.** To examine genetic interactions among the four dTFs, we analyzed the transcriptomes of several dual dTF-cKOs: Ptf1a with Nr5a2, Foxa2 and Gata4, as well as Gata4 with Nr5a2 and Foxa2. All but the Foxa2+Gata4-cKO mice had largely additive effects on PAC gene expression (**Fig 6A**); that is, 60–75% of the affected genes of the two individual dTF-cKOs were present in the dual cKO, and additional affected genes not affected in the single cKOs were less than 36%. However, the Foxa2+Gata4-cKO had a much greater than additive effect: fully 63% of the DE genes were not affected in either single cKO (**Fig 6B**). This appears due in large part to the loss of *Ptf1a* expression in the double Foxa2+-Gata4-cKO compared to only modest (and opposite) effects in the individual cKOs (**Fig 6C**). As a result, nearly 80% of Ptf1a-cKO affected genes were altered in the dual cKO compared to the single cKOs (**Fig 6D**), and additional cellular pathways enriched for affected genes of the double-cKO were derived from Ptf1a target genes (**Fig 6E**). Either Foxa2 or Gata4 are critical for continued *Ptf1a* expression and therefore the maintenance of acinar differentiation.

Although all four dTFs bind a proven enhancer of each other's genes (**Fig 2H**), many fewer genuine regulatory effects were confirmed by the mRNA analyses of the cKOs (**Fig 6F**). Ptf1a activates *Nr5a2* and autoactivates its own gene through the same ARD [98] also bound (**Fig 6G**) and regulated by Foxa2 and Gata4. Nr5a2 and Gata4 appear to auto-suppress (**Fig 6F**), because their inactivation increased the levels of their residual mRNAs. The four dTFs bind and regulate at least 178 genes encoding DNA-binding TFs (S2 Fig).

Two key downstream transcription factors, Mist1 and Rbpjl, are controlled by Ptf1a, Nr5a2 and Foxa2 (**Fig 6F**). Mist1 and Rbpjl are highly expressed, cell-restricted, "scaling" transcription factors of late acinar differentiation [17,99]. Scaling factors act downstream of dTFs to coordinate the optimization of high-capacity exocytosis in select secretory cell-types [100,101]. Though important to complete and maintain PAC differentiation and homeostasis, Mist1 and Rbpjl play no role in specifying or initiating the acinar differentiation program but are the most important downstream transcriptional effectors of the dTFs [17,99,102]. The four dTFs maintain the PAC transcriptional program through extensive co-regulation of the genes for acinar transcription factors.

**Co-regulation of the Mist1 enhancer by the dTFs.** To examine whether binding of each dTF is required for co-regulation of an PAC-specific transcriptional enhancer, we analyzed the effects of mutating dTF-binding sites on the activity of the *Mist1/Bhlha15* ARD in vivo. The *Mist1* ARD binds all four dTFs (**Fig 7A**), and the Mist1 mRNA level is affected by Ptf1a, Foxa2 and Nr5a2 (**Fig 7B**). The 709-bp ARD (**Fig 7C**) linked to a *lacZ*-reporter was activated in acinar tissue in a high fraction of founder transgenic mice (**Fig 7D, 7F and 7G**). Previous work had shown that mutations that disrupt binding of the PTF1 complex to the *Mist1* ARD also disrupted ARD activity in transgenic mice [99], consistent with dependence on Ptf1a and

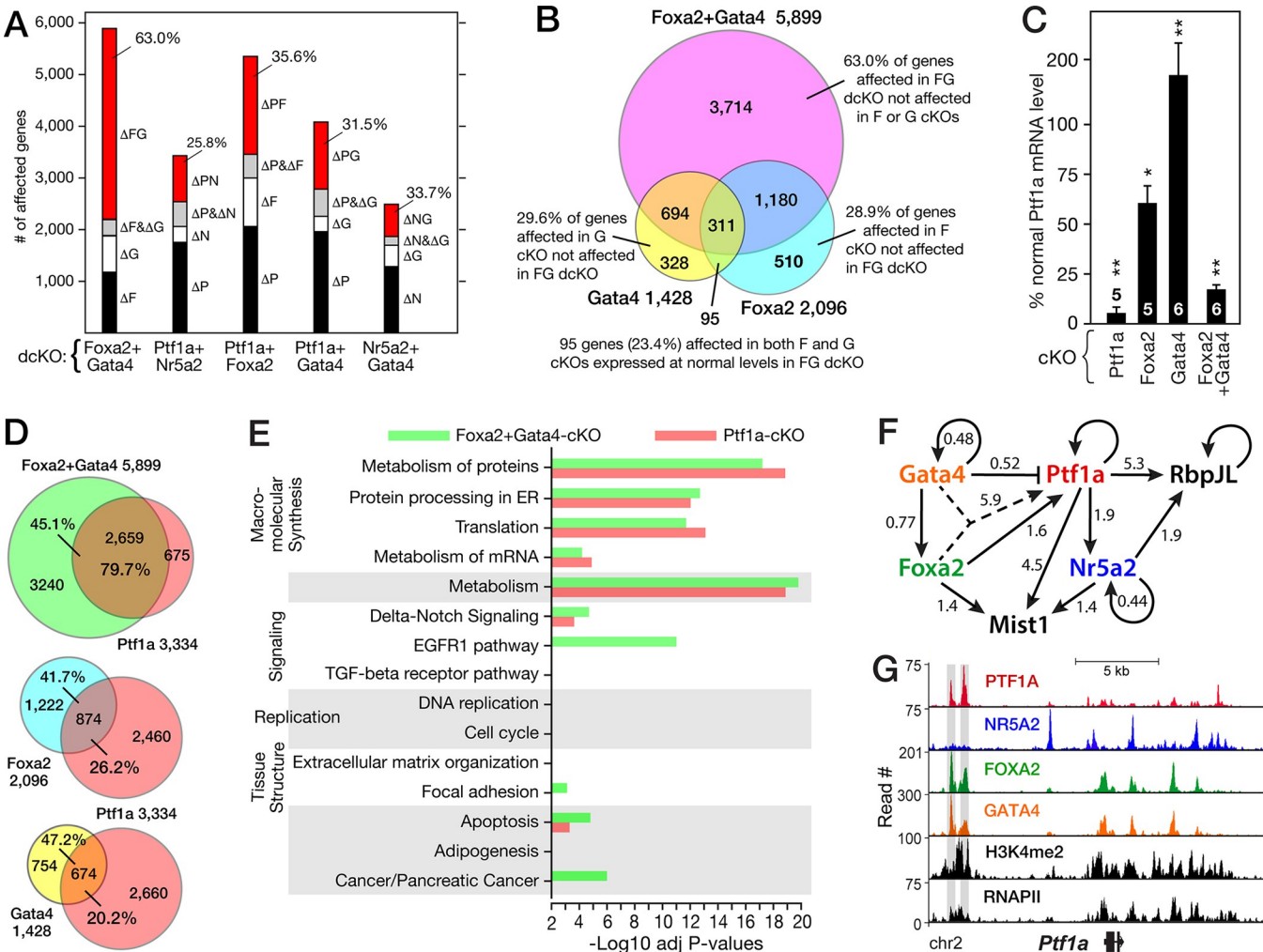

**Fig 6. Effects of dual dTF-knockouts on PAC gene expression. A**. Quantification of double-cKO affected genes restricted to one (*black*) or the other (*open*) single cKO, or to both (*grey*), and additional genes not affected in either single cKO (*red*). % indicates the 'more than additive' fraction for each double cKO. **B**. Overlap for sets of DE genes for Foxa2-, Gata4- and Foxa2+Gata4-cKOs. **C**. Effects of *Foxa2* and *Gata4* inactivation on Ptf1a mRNA levels. **D**. Most of the affected genes in the Ptf1a-cKO (*red*) are also affected in the Foxa2+Gata4-cKO (green), but not in either Foxa2 (blue) or Gata4 (yellow) single cKOs. **E**. Enriched cellular processes for the 5,989 Foxa2+Gata4-cKO affected genes (*green*) are dominated by those for the Ptf1a-cKO (*red*) and distinct from those of Foxa2- and Gata4-cKOs (compare Fig 1F). **F**. Core dTF architecture of cross-regulation based on dTF binding and regulatory contributions revealed by the cKOs. The values quantify the effect of the dTF on the level of expression of the target (the inverse of effect by the cKO). Note that the Foxa2-Gata4 double cKO reveals that *Ptf1a* expression requires either Foxa2 or Gata4 (*dashed arrow*). **G**. Binding of the dTFs at the ARD (*shaded bars*) of *Ptf1a*.

demonstrating the efficacy of this in vivo assay. To test for a direct requirement for the binding of the other dTFs, we identified and mutated (**Fig 7E**) all four recognized binding sites for Nr5a2, five for Foxa2, or four for Gata4 within the dTF-bound regions of the ARD (**Fig 7D**). Mutational inactivation of the Nr5a2-binding sites had no discernible effect in transgenic mice (**Fig 7D**), despite the partial dependence of *Mist1* expression on *Nr5a2* (**Fig 7B**). Moreover, even though the Foxa2-cKO only modestly affected and the Gata4-cKO slightly increased *Mist1* expression, mutation of all Foxa2 binding sites or all Gata4 sites completely inactivated the enhancer. These results suggest that Foxa- and Gata-protein binding are crucial to activity of the enhancer, and that other Foxa and Gata paralogs may partly compensate in this instance for the absence of Foxa2 and Gata4, respectively. Indeed, *Gata6* expression increases 2-fold in

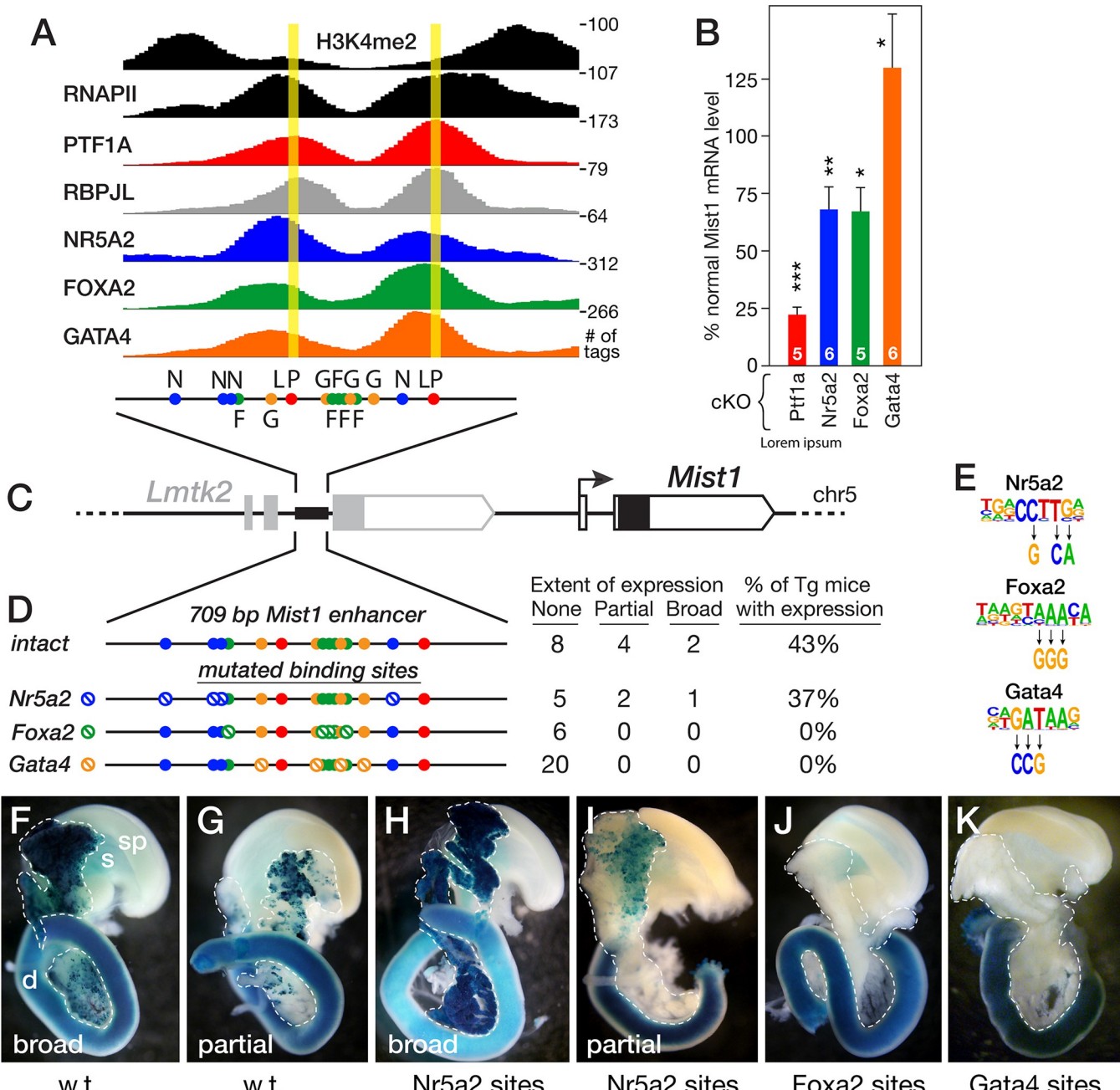

**Fig 7. Regulation of the *Mist1/Bhlha15* transcriptional enhancer. A**. Tag-distribution at the *Mist1* enhancer from ChIP-Seq analyses of RNAPII, H3K3me2 and the dTFs. The positions of consensus binding sequences (*filled circles*) for the dTFs are shown in the 709 bp enhancer (colors correspond to those of the ChIP-seq peaks). *yellow lines*, pinnacles of the Ptf1a peaks. **B**. Effects of the cKO for each dTF gene on the level of Mist1 mRNA from RNA-Seq data. Numbers in bars, number of mice; t test p values: *<0.05, **<0.01, *** <0.001 vs. normal. **C**. Diagram of the *Mist1* locus including the *Mist1* enhancer within the last intron of the *Lmtk2* gene. **D**. Results for the in vivo activity of the *Mist1* enhancer; altered binding sequences (panel E) are indicated by stop icons, with numbers of mice with no, partial, or broad pancreatic expression. **E**. Mutations used in consensus binding sequences that abolish binding of the cognate dTFs. **F**-**K**. Importance of the dTFs for enhancer activity during late fetal development (E17.5). Representative examples of the beta-galactosidase-staining for enhancer-driven lacZ-transgenes with the unmodified (*w.t.*) enhancer and versions with multiple disrupted binding sequences of Nr5a2, Foxa2, or Gata4. The dashed lines delimit pancreatic tissue. *sp*, spleen; *s*, stomach; *d*, duodenum; staining of the duodenum is due to endogenous beta-galactosidase, not the transgene.

the Gata4-cKO, and Gata6 plays an important role in acinar maintenance [46]; no *Foxa* para-log increased in the Foxa2-cKO.

**Resistance to Kras-driven PAC transformation.** Because caerulein-treatment induces de-differentiation and increases sensitivity to oncogenic Kras in mouse models [20,103,104], we examined whether the loss of acinar differentiation due to dTF gene inactivation also enhanced the effects of oncogenic Kras (**Fig 8**). To best recapitulate human PDAC initiation, we induced expression of oncogenic $Kras^{G12D}$ in adult mouse pancreas [104–106], and simultaneously inactivated one or two dTF genes. Expression of $Kras^{G12D}$ alone induced acinar-to-ductal metaplasia (ADM) in a small subset of acinar cells by 12 weeks and PanINs at about 20 weeks (**Fig 8A**). Homozygous inactivation of either *Ptf1a* or *Nr5a2* simultaneously with induction of $Kras^{G12D}$ greatly accelerated the appearance and extent of ADM to 1–2 weeks and PanINs to 2–4 weeks. Localized ADM was characterized by the loss of acinar morphology (**Fig 8C**), decreased expression of the digestive enzyme genes, and the induction of pancreatic ductal cell markers Sox9 and Ck19/Krt19 [107] (**Fig 8I** and S7 Fig). Extensive regions of advanced ADM (**Fig 8D**) were distinguished by the absence of Ptf1a, decreased Nr5a2, and the appearance of low levels of the PanIN-marker Cldn18 [108], the transformation marker phospho-MAPK, and tuft cell markers Dclk1 and Vav1 (**Fig 8I**). PanINs (**Fig 8E and 8F**) were identified by high Cldn18, Muc5ac, Alcian blue staining of acidic mucins, and phospho-MAPK as well as increased numbers of Dclk1+ and Vav1+ tuft cells (**Fig 8I, 8O and 8P** and S7 Fig). Lineage tracing from the onset of $Kras^{G12D}$ induction using a co-induced tdTomato marker showed that the epithelial cells of ADM, PanIN and AFL lesions derived from adult PACs (**Fig 8I** and S7 Fig). However, tuft cells in PanIN epithelia did not contain the PAC lineage marker (**Fig 8P**), in contrast to the results of a previous report that included caerulein-induced ADM in combination with $Kras^{G12D}$ transformation [109].

$Kras^{G12D}$ expression in the absence of Ptf1a or Nr5a2 in nearly all acinar cells lead to the loss of 80–93% or 50–80%, respectively, of pancreatic mass at 4 weeks (**Fig 8J**). The remaining tissue comprised PanINs, ductal structures, intact islets, and fibrotic tissue. In contrast, the absence of Foxa2 or Gata4 did not lead to $Kras^{G12D}$-induced cell loss nor to PanINs for at least 19 months. Because PanINs were plentiful at 20 weeks for $Kras^{G12D}$ without dTF depletion, it appears that Foxa2 and Gata4 are required for $Kras^{G12D}$-mediated PanIN formation.

We note that other acinar lesions previously proposed to be potential preneoplastic lesions were also observed in pancreas with induced $Kras^{G12D}$ and depleted dTFs (**Fig 8A**). Atypical Flat Lesions (AFL) [110,111] were present in Foxa2- or Gata4-depleted $Kras^{G12D}$ pancreas and increased in double Foxa2+Gata4-depleted pancreas with $Kras^{G12D}$ (**Fig 8G**). Ductular-Insular Complexes (DICs) (**Fig 8H**) are PanIN-like epithelial structures [112] that stain with Alcian-blue (**Fig 8K and 8L**), contain Dclk1-positive tuft cells (**Fig 8O**), and include discrete islet-like, synaptophysin- and insulin-positive endocrine cell clusters (**Fig 8M and 8N**). DICs were present in all genotypes of $Kras^{G12D}$-expressing pancreas except those depleted for Foxa2 and/or Gata4.

To determine whether $Kras^{G12D}$-mediated transformation depends on the level of acinar differentiation, we first calculated a quantitative PAC deDifferentiation Index (dDI) using the number of genes affected for five PAC differentiation programs for each dTF-cKO (S6 Table). No loss of differentiation is a dDI score of zero, and higher numbers infer greater loss. The dDI scores for the cKOs were *Foxa2+Gata4*, 357; *Ptf1a*, 324; *Nr5a2*, 133; *Foxa2*, 130; *Gata4*, 64 (**Fig 8Q**). These scores as well as qualitative assessment of acinar morphology showed that the *Foxa2+Gata4* double cKO was at least as detrimental to PAC differentiation as the Ptf1a-cKO and more so than the Nr5a2-cKO, both of which greatly amplified the response to $Kras^{G12D}$. If simply the magnitude of dedifferentiation was key to efficient $Kras^{G12D}$-transformation of acinar cells, then the *Foxa2+Gata4* double cKO should rapidly generate $Kras^{G12D}$-induced

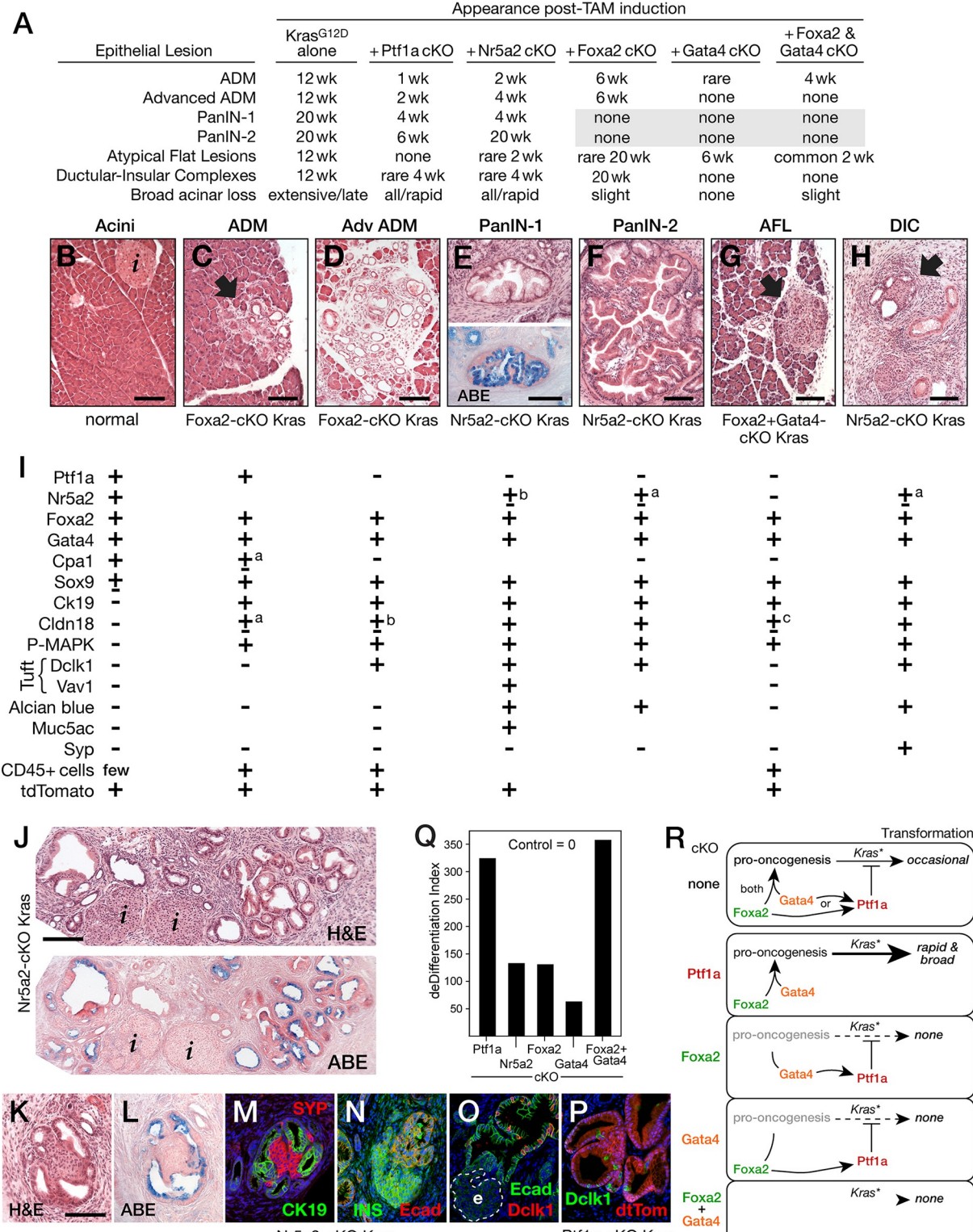

**Fig 8. Effects of the loss of dTFs on *Kras*^G12D^ transformation of pancreatic acinar cells in situ. A**. Timing of appearance of preneoplastic lesions in *Kras*^G12D^-expressing pancreas with dTF gene inactivations or without. *Grey shading,* PanINs suppressed. **B-H**. Representative examples of preneoplastic lesions (genotypes below). *Filled arrows* indicate discrete lesions; *i*, islet. **B**. Acinar rich normal pancreas. **C**. Acinar-to-Ductal Metaplasia (ADM). **D**. Advanced ADM. **E**. PanIN-1; lower image is Alcian blue + eosin (ABE) staining of the same PanIN in a nearby section. **F**. PanIN-2. **G**. Atypical Flat Lesion (AFL). **H**. Ductular-insular Complex (DIC). **I**. Association of acinar and lesion markers

with normal tissue and lesions. *tdTomato*, acinar lineage trace using *Rosa-CAG-LSL-tdTomato*. Footnotes: [a] staining of a minority of epithelial cells in lesions and at low intensity; [b] majority/low; [c] majority/high. **J**. Low magnification across sections showing complete absence of acini, extensive ADM and PanIN (similar for Nr5a2- and Ptf1a-deficiency). **K-M**. Nearby sections of a DIC with H&E (**K**), ABE (**L**), and immunodetection of Ck19 and synaptophysin plus nuclear Dapi (**M**). **N**. Insulin immunostaining of the endocrine core of a DIC. **O**. Dclk1-positive tuft cells in DIC epithelium; *e*, endocrine cluster. **P**. tdTomato lineage-trace does not mark Dclk1-positive tuft cells. Size bars, 100 m. **Q**. deDifferentiation indices for the cKOs. **R**. Scheme for the regulatory relationships among *Foxa2*, *Gata4* and *Ptf1a* and their effects on acinar transformation by *Kras^{G12D}* (*Kras**). See text for explanation.

PanIN formation and acinar cell loss. Instead, PanINs did not form, and acinar cell loss did not occur. Therefore, Ptf1a and Nr5a2 must maintain expression of yet unidentified effectors resisting the action of genes that promote transformation by oncogenic Kras.

## Discussion

In this report we delineate the actions of four differentiation transcription factors that maintain the highly differentiated state of the pancreatic acinar cell, including the massive production and secretion of digestive hydrolytic enzymes, a specialized anabolic metabolism, proteostasis, replicative quiescence, and resistance to oncogenic transformation. Each dTF affected expression of the secretory protein genes, the definitive markers of PACs that account for approximately 90% of acinar protein synthesis. In addition, Ptf1a affects all aspects of protein production and acinar anabolic metabolism broadly; Nr5a2 and Foxa2 affect genes of replication, metabolism and intercellular signaling; Foxa2 also the extracellular matrix; and Gata4 principally affects genes of metabolism and secretory protein processing. Whereas pairwise inactivations of dTF genes generally caused simple additive effects on the number of affected genes, the disruption of *Foxa2* and *Gata4* together had a more than additive effect, due in large part to greatly decreased *Ptf1a* expression specific to this dual cKO. The analysis of Foxa2+Gata4-cKO also uncovered a novel genetic interaction in which repressive action by Foxa2 on replication genes is dependent on the presence of Gata4.

The far-reaching control by the four dTFs is reinforced by forward auto-regulation of the *Ptf1a*, *Nr5a2* and *Gata4* genes, maintenance of *Nr5a2* transcription by Ptf1a, and cooperative activation of *Ptf1a* by Foxa2 and Gata4 (**Fig 6F**). *Ptf1a* transcription is further ensured by the product of one of its direct target genes, Rbpjl, which is incorporated in the functional trimeric complex of PTF1-L [12]. The dTFs effect downstream gene control through direct and overlapping regulation of at least 178 DNA-binding TFs (S2 Fig) and in likely partnership with several transcription cofactors including Nfi, Sp1/2, Hnf1b, Atf1, Elk1, the co-repressor complex Thap11:Zfp143.

### Acinar protein synthesis

The defining feature of the exocrine pancreas is the prodigious production of digestive enzymes equal to mass of the gland every twenty-four hours. Depending upon diet, one-quarter to one-third of the amino acids absorbed by the human intestine derive from the auto-hydrolytic digestion of the pancreatic enzymes themselves [113]. The four highly expressed acinar differentiation transcription factors maintain the unrivaled high rate of PAC protein synthesis and the metabolism that powers it.

Four common processes control the rate of cellular protein synthesis.

The first is the amount of translational machinery (i.e., ribosomes) and its substrates (e.g., amino acids and acylated tRNAs). The murine pancreas has the most ribosomes per cell and the highest protein synthetic rates of somatic tissues [114,115]. Ptf1a, Nr5a2, Foxa2 and Gata4 bind and activate genes for the brick-and-mortar translational machinery. Ptf1a and Foxa2 +Gata4 also maintain the expression of *Myc*, which plays a prominent support role for the high protein synthetic capacity of PACs [116].

The second strategy is to control the total amount of mRNA. Murine PACs have 0.5–1 million mRNA molecules per cell [115]; of those about 95% encode the secretory enzyme mRNAs [77] and account for at least 90% of acinar protein synthesis [117]. The effects of the cKOs show that the four dTFs drive the extraordinarily high accumulation of the secretory protein mRNAs. Within six days, inactivation of *Ptf1a*, *Foxa2*, and *Gata4* decreased acinar mRNA levels by 33, 39 and 12%, respectively (see S6A Fig).

The third is through translational regulators, principally specific protein kinases as well as their translation initiation factor targets. Inactivation of the dTF genes decreased the mRNAs of regulatory proteins known to be crucial [80] for optimized pancreatic acinar protein synthesis (**Fig 3K**). For example, decreased levels of phosphorylation of Rps6 and the regulatory EIF4F scaffolding subunit Eif4g correlated with the decreased levels of the mRNAs for their respective protein kinases, Akt1/Rptor/Rps6kb1 and Mknk1.

A final means is the potent anabolic metabolism that powers acinar protein biosynthesis. The inactivation of *Ptf1a*, *Nr5a2*, *Foxa2* and *Gata4* affected the mRNA levels of 34.3, 16.3, 21.6, and 12.8 percent, respectively, of the 1386 KEGG-listed metabolism genes (S2 Table) expressed in the pancreas. The dTFs affected intermediary metabolism, especially the enhanced expression of genes for the metabolism of amino acids, glutamine-uptake and entry into the TCA cycle, the methionine cycle, and creatine biosynthesis. The supercharged anabolic metabolism is a vital property of the highly differentiated state of PACs.

The manufacture of creatine and its precursor guanidinoacetate at high levels is a specialized function of PACs [95,118,119]. The phosphorylated form of creatine is the major source of energy that replenishes ATP consumed during prolific zymogen granule exocytosis at the luminal membrane [96]. In addition, excess guanidinoacetate released by PACs into the portal vein supports the hepatic biosynthesis of creatine for muscle use [119]. Gamt is the methyl transferase that converts guanidinoacetate to creatine by methylating the d-Nitrogen of the guanidino group. In the pancreas, Gamt is restricted to acinar cells, and we showed that it is present at high levels in the nucleus and cytoplasm. A potential yeast orthologue, Rmt2p, is nuclear and has the identical specificity for methylating the d-N of arginine residues in proteins [120]. All other known protein arginine methyltransferases methylate only the terminal w-nitrogen of the guanidino side chain [121]. The presence of Gamt in the nucleus is an intriguing observation that suggests a novel function in acinar cell nuclei.

## Acinar proteostasis

Pancreatic acinar cell homeostasis is fragile due to the profusion of misfolded protein [53] that accrues from the enormous production of secretory enzymes. Acinar proteostasis is sustained principally by a highly tuned Unfolded Protein Response driven by *Ptf1a*-control [18] and supported by the autocrine/paracrine effects of Fgf21 [89]. *Fgf21* expression is as selective as the digestive enzyme genes and highest in the pancreas [90]. The acinar transcription of *Fgf21* is specified by an ARD bound and regulated by Ptf1a, Foxa2 and Gata4; consequently, some of the deleterious effects of the dTF-cKOs may be due to diminished autocrine signaling by Fgf21. *Fgf21* expression disappears in the dedifferentiated acinar tissue of induced-pancreatitis, and administration of Fgf21 restores pancreatic levels and acinar function dramatically [90]. In this regard, Fgf21 may provide an effective treatment for pancreatitis.

### Functional analysis of a PAC-restricted transcriptional enhancer

Mist1 is a key downstream effector controlled by Ptf1a, Nr5a2 and Foxa2. To test whether the effects of the dTF-cKOs on *Mist1* expression correlate with a requirement for dTF binding to the *Mist1* ARD, we measured the consequence of inactivating all recognizable binding sites for

Nr5a2, Foxa2 and Gata4 in the enhancer. The results for the *in vivo* analyses confirmed functional roles for Foxa2 and Gata4 binding, as previously for Ptf1a and its dedicated cofactor Rbpjl [99]–although this transgenic analysis cannot distinguish a requirement for activation of the enhancer during pancreatic development versus maintenance of transcription in adult acinar cells.

Contrary to expectations, we found that mutational inactivation of the Nr5a2 binding sites had no discernible effect on enhancer activity in vivo, even though the Nr5a2-cKO decreased *Mist1* expression severely. The absence of an effect has several possible interpretations: *i*, the retention of unidentified, cryptic binding sites for Nr5a2 in the mutated enhancer; *ii*, a nonredundant enhancer in the *Mist1* locus that is Nr5a2-dependent; and *iii*, nonfunctional Nr5a2 binding. This last possibility might be an example of lineage specific TFs, present at very high nuclear concentrations, binding regions of open chromatin such as ARDs irrespective of function at some sites [65]. These complications illustrate the limitations to reconciling the behavior of isolated enhancers, which may represent only part of a gene's regulatory information, with the transcriptional behavior of the entire gene in situ.

## Foxa2 and Gata4 collaboration

The results of the transcriptomic analyses of the single cKOs for Foxa2 and Gata4 and the compound Foxa2+Gata4-cKO revealed a greater-than-additive effect for the dual cKO and a novel collaboration between Foxa2 and Gata4 at acinar replication genes. The greater than additive effect was principally due to the dramatic decreased expression of *Ptf1a* in the dual cKO. Thus, continued expression of *Ptf1a* as well as its downstream targets requires either Foxa2 or Gata4. In this regard, the requirement of Foxa2 or Gata4 for the expression of the key acinar scaling factor *Mist1* appears to be indirect through their maintenance of *Ptf1a*.

Acinar replication and many genes controlling replication were induced upon depletion of Foxa2 but were unaffected by depletion of Gata4 or the dual depletion of Foxa2 and Gata4. These results are consistent with Foxa2-mediated repression that is dependent on the presence of an activating Gata4 at promoters of regulated genes (**Fig 4K**). This establishes the importance of Foxa2 and Gata4 for acinar replication as well as a new regulatory mechanism by Foxa2 and Gata4 collaboration. Although other Foxa- (i.e., Foxa1/3) and Gata4-like (i.e., Gata6) factors are present in acinar cells, they appear unable to substitute for the Foxa2 and Gata4 regulatory interactions at genes for DNA replication and for the maintenance of *Ptf1a* expression. Because intestine, colon, stomach and liver also co-express *Foxa2* and *Gata4*, their relationship controlling replication may operate in these epithelial glands as well.

## The dTFs affect oncogenic transformation

The opportunity to activate expression of oncogenic $Kras^{G12D}$ selectively in the acinar cells of adult mice led us to investigate the roles of the four dTFs in a genetic model of oncogenic transformation that mimics the initiation of human pancreatic cancer *in vivo*. Pharmacological doses of caerulein, a potent analogue of the secretagogue cholecystokinin, induce acinar dedifferentiation, incipient inflammation, and greatly heightened sensitivity to Kras transformation that leads to pancreatic tumors in mice [60]. We initially wanted to test whether the enhanced susceptibility to $Kras^{G12D}$ transformation associated with caerulein-induced pancreatitis and *Ptf1a* inactivation [20] was due to sudden acinar dedifferentiation. The other dTFs have quantitatively different effects on differentiation, and we found by comparison that the severity of dedifferentiation of the cKOs (Foxa2+Gata4 > Ptf1a > Nr5a2 > Foxa2 > Gata4) did not, in fact, correlate with their effects on $Kras^{G12D}$-driven transformation. The rapid induction of widespread Kras-induced acinar cell loss suggests that the absence of Ptf1a or Nr5a2

supercharges Kras activity by enhanced GTP-loading. However, even though inactivation of *Foxa2*, *Gata4*, or *Foxa2*+*Gata4* caused extensive loss of differentiation, none promoted cell-loss and prevented rather than promoted transformation. In this *in vivo* model, Ptf1a and Nr5a2 are tumor suppressive, whereas Foxa2 and Gata4 are pro-oncogenic. We note that the other Foxa and Gata paralogs present in acinar cells do not compensate for the pro-oncogenic activity of Foxa2 and Gata4. The contrasting effects of the dTF-cKOs on acinar transformation are most likely due to altered expression of distinct effector genes, some perhaps shared by the Ptf1a- and Nr5a2-cKOs and others by the Foxa2- and Gata4-cKOs, rather than overall quantitative effects on PAC differentiation status.

The scheme presented in **Fig 8R** interprets the effects of the conditional inactivations of *Foxa2*, *Gata4* and *Ptf1a* on each other and on the ability of oncogenic Kras to transform adult PACs. A pro-oncogenic state requiring *Foxa2* and *Gata4* supports transformation by oncogenic $Kras^{G12D}$ and *Ptf1a* limits this permissiveness to a low level. Although *Ptf1a* has little control over *Foxa2* or *Gata4* expression, *Ptf1a* expression requires either *Foxa2* or *Gata4* (**Fig 6F**). Even though *Ptf1a* expression is lost in the combined Foxa2+Gata4-cKO, $Kras^{G12D}$-driven transformation does not occur. This suggests that the pro-oncogenic state in the Foxa2+-Gata4-cKO is depleted to the extent that *Ptf1a* is not needed to suppress transformation. Because acinar cells are the predominant pancreatic cell type and dTF gene inactivation and $Kras^{G12D}$ induction occur concertedly in nearly all acinar cells, biochemical approaches using the entire pancreas can resolve acinar transcriptomic changes, chromatin modifications, and proteomic alterations *in vivo*. Such an approach now has the potential to identify relevant transforming cofactors, affected signaling pathways, and cellular processes complicit with oncogenic Kras and encouraged by pro-oncogenic *Foxa2* and *Gata4* and antagonized by *Ptf1a* and *Nr5a2*.

## Materials and methods

### Mouse genotypes and treatment

Mice used in this study with the following alleles have been described: $Ptf1a^{CreERT}$ (Ptf1a^{tm2(cre/ESR1)Cvw}) in which most of *Ptf1a* exon 1 is replaced by the CreERT coding region [68]; $Ptf1a^{fl}$ (Ptf1a^{tm3Cvw}) [20]; $Nr5a2^{fl}$ (Nr5a2^{tm1Sakl}) [26]; $Foxa2^{fl}$ (Foxa2^{tm1Khk}) [122]; and $Gata4^{fl}$ (Gata4^{tm1.1Sad}) [123]. The complex genotypes derived from these strains are described in the table below. For the inactivation of individual dTF genes, mice were treated with 0.25 mg of tamoxifen per gram weight by oral gavage at 3–4 pm for three consecutive days (days 0, 1 and 2) and pancreases obtained by dissection on day 6. The *Ptf1a* heterozygote $Ptf1a^{CreERT/+}$ (normal) was used as for the control pancreas because all dTF-cKO genotypes included the $Ptf1a^{CreERT}$ allele. The tdTomato lineage trace allele was derived from $ROSA26Sortm14^{(CG-tdTomato)}{}^{Hze}$ mice [124]. Transgenic mouse embryos for the in vivo *Mist1* enhancer analyses were generated as previously described [99]. All experiments involving mice were performed according to NIH guidelines and approved by the Institutional Animal Care and Use Committee of the University of Texas Southwestern Medical Center.

| Mouse pancreases analyzed for each dTF genotype | | |
|---|---|---|
| **Shorthand designation** | **genotype** | **# pancreases for RNA-seq** |
| normal (control) | $Ptf1a^{CreERT/+}$ | 8 |
| Ptf1a-cKO | $Ptf1a^{CreERT/fl}$ | 5 |
| Nr5a2-cKO | $Nr5a2^{fl/fl}$; $Ptf1a^{CreERT/+}$ | 6 |

*(Continued)*

**Mouse pancreases analyzed for each dTF genotype**

| Shorthand designation | genotype | # pancreases for RNA-seq |
|---|---|---|
| Foxa2-cKO | $Foxa2^{fl/fl}$; $Ptf1a^{CreERT/+}$ | 5 |
| Gata4-cKO | $Gata4^{fl/fl}$; $Ptf1a^{CreERT/+}$ | 6 |
| Foxa2+Gata4-cKO | $Foxa2^{fl/fl}$; $Gata4^{fl/fl}$; $Ptf1a^{CreERT/+}$ | 6 |
| Kras$^{G12D}$ | $Kras^{LSL-G12D/+}$; $Ptf1a^{CreERT/+}$ | |
| Ptf1a-cKO Kras | $Ptf1a^{CreERT/fl}$; $Kras^{LSL-G12D/+}$ | |
| Nr5a2-cKO Kras | $Kras^{LSL-G12D/+}$; $Nr5a2^{fl/fl}$; $Ptf1a^{CreERT/+}$ | |
| Foxa2-cKO Kras | $Kras^{LSL-G12D/+}$; $Foxa2^{fl/fl}$; $Ptf1a^{CreERT/+}$ | |
| Gata4-cKO Kras | $Kras^{LSL-G12D/+}$; $Gata4^{fl/fl}$; $Ptf1a^{CreERT/+}$ | |
| Foxa2+Gata4-cKO Kras | $Kras^{LSL-G12D/+}$; $Foxa2^{fl/fl}$; $Gata4^{fl/fl}$; $Ptf1a^{CreERT/+}$ | |
| Kras lineage-trace | $Kras^{LSL-G12D/+}$; $Ptf1a^{CreERT/+}$; $Rosa26^{CAG-LSL-tdTomato/+}$ | |

## RNA-seq and data analysis

We refined a previous RNA-seq analysis for pancreatic inactivation of *Ptf1a* [18] by increasing the number of RNA-seq data sets for individual Ptf1a-cKO mice ($Ptf1a^{fl/CreERT}$) from three to five and for control mice ($Ptf1a^{CreERT/+}$) from three to eight. The results from the expanded Ptf1a-cKO analysis confirmed all prior conclusions [18]. The bioinformatic analysis of the *Nr5a2* conditional inactivation was improved over a prior analysis of whole body gene inactivation [30] by restricting gene disruption to pancreatic acinar cells using Ptf1a-CreERT (genotype $Ptf1a^{CreERT/+}$; $Nr5a2^{fl/fl}$) and by increasing the number of Nr5a2-cKO pancreases from three previously analyzed as a single pool for RNA-seq to six analyzed individually. The new Ptf1a- and Nr5a2-cKO data compilations were analyzed with parameters identical to those for the Foxa2-cKO and Gata4-cKO (see Methods) to allow direct comparisons (see below). The enhancements derived from the greater sample sizes identified more differentially expressed (DE) genes with high confidence and enriched cellular pathways while retaining the selective emphasis on the previously identified PAC processes (S7 Table).

For total RNA isolation, adult mouse pancreases were collected at 3–4 pm and processed using the guanidium thiocyanate procedure [125] with minor modifications [18]. Separate amplified cDNA libraries for each pancreas were prepared for RNA-seq and sequenced as previously described [18]. Sequencing data were processed and analyzed by edgeR [70] as described in reference [18] as a consideration of the extreme bias of the acinar pancreas mRNA population with extraordinarily high levels of the approximately two dozen mRNAs for the digestive enzymes [77] affected by the dTF conditional knockouts. The RNA-seq data sets for the Ptf1a-, Nr5a2-, Foxa2- and Gata4-cKOs as well as the normal control pancreases are accounted in the GEO data set GSE100881.

## ChIP-seq and data analysis

Preparation of chromatin from mouse pancreas, chromatin immunoprecipitation, preparation of the amplified ChIP-seq libraries for sequencing, and analysis of the resulting data sets were performed as described previously [18]. Separate ChIP-seq experiments were performed with two independently derived antibodies for each dTF. dTF binding peaks were called using HOMER [126] with an fdr of <0.01; consensus peaks were identified using intersectBed. dTF-bound sites were associated with nearby genes using GREAT with a basal 5+1-kb and extension to 1Mb [73]. The ChIP-seq data sets for Ptf1a, Nr5a2, Foxa2 and Gata4 are available

under GEO accession numbers GSE86262 and GSE100881. Data sets for H3K4me2 and RNA polymerase II binding are in GSE86289.

Further analyses of the ChIP-seq peaks were performed with the HOMER software suite [126]. Active Regulatory Domains (ARDs) are defined as H3K4me2-containing regions (HOMER histone region parameters S500-M5000) containing a peak(s) for RNA polymerase II (RNAPII); the median ARD length was 1329-bp. Colocalization of dTF binding defined by overlapping ChIP-seq peaks with peak widths standardized to 200 bp. Oligonucleotide sequence enrichment of consensus binding sites in peaks was determined with HOMER findMotifsGenome.

## Measurement of the rate of protein synthesis

Pancreases were collected from mice 6 days after the initiation of tamoxifen treatment at 9–10 am, dissected in cold F12 medium (plus 10 mM glutamine, 1% bovine serum albumin, and 0.1 mg/mL soybean trypsin inhibitor) to obtain individual lobules not larger than 1 x 2 x 0.5 mm [127]. Ten lobules from each pancreas were incubated in 5 mL of the modified F12 medium with 25 μCi of $^{35}$S-methionine/cysteine (EXPRESS35S, PerkinElmer) for 10, 20 and 40 min with 100% oxygen flow at 0.2 cubic feet per min and continuous shaking [128]. The incubated lobules were homogenized with an Evolution bead homogenizer (Precellys). Two aliquots of each homogenate were treated with 10% trichloroacetic acid to precipitate protein, rinsed with 10% trichloroacetic acid, then dissolved in 0.1 N NaOH prior to measuring incorporated $^{35}$S by scintillation counting. To correct for relative cell-number, DNA was measured from a sample of each homogenate using the QuantiFluor system (Promega). The protein synthesis rates were expressed as $^{35}$S-cpm/μg DNA/min.

## Histology, immunodetection and image analyses

Adult pancreas was fixed for histology with 4% paraformaldehyde, 0.1 M sodium phosphate buffer, pH 7.4 at 4˚C overnight and embedded in paraffin or cryo-embedded in OCT following established protocols [18,24]. Sections for immunofluorescence were pretreated at high temperature and pressure (Retriever 2100, Aptum Biologics) with Antigen Unmasking Solution (Vector) for antigen retrieval. Micrographic images are representative of at least three animals for each genotype. The sources and dilutions for the antibodies used for immunofluorescence are listed in S8 Table.

Acinar cells in situ were lineage-traced by tamoxifen-induced CreERT-mediated recombination of *Rosa-CAG-LSL-tdTomato* locus (MGI:3809524) (124). For mice also bearing *Ptf1a-CreERT* and floxed dTF genes, the tamoxifen treatment coincidentally activated tdTomato expression, induced *Kras$^{G12D}$*, and inactivated the dTF genes. Cell-specific expression of tdTomato was determined with morphologic criteria and by co-immunofluorescent staining with lesion markers (S8 Table).

## Accession numbers

RNA-seq and ChIP-seq data sets from this work were deposited in the GEO database under SuperSeries accession numbers GSE10088, GSE86290 and GSE100881.

## Supporting information

**S1 Fig. Decision tree to identify the four differentiation TFs and the flow/rationale for analyses of the roles they play in key cellular processes that define acinar differentiation status and identity.**
(PDF)

**S2 Fig. The 70 highest expressed transcription factors for the mouse pancreas, parotid and liver.**
(PDF)

**S3 Fig. Regulatory interactions among the dTFs and their control of genes encoding other DNA-binding sequence-specific transcription factors.**
(PDF)

**S4 Fig. The sets of signature genes for each dTF cKO confirm the selective regulatory responsibilities of the individual dTFs.**
(PDF)

**S5 Fig. Correlation of dTF binding to active enhancers with the effects of dTF-cKOs.**
(PDF)

**S6 Fig. Pathway enrichment for genes with ARDs bound by dTFs and potential cofactors.**
(PDF)

**S7 Fig. Regulation of protein synthesis.**
(PDF)

**S8 Fig. Additional distinguishing immunofluorescent markers of preneoplastic lesions.**
(PDF)

**S1 Table. Efficiency of floxed-dTF deletion by Ptf1a$^{CreERT}$.**
(PDF)

**S2 Table. Gene sets for pathways in text Figs 1, 3, 4 and 5.**
(DOC)

**S3 Table. RNAseq results: All cKOs, all expressed genes.**
(XLSX)

**S4 Table. Fourteen stomach- or intestine-restricted genes affected by dTF-cKOs and indicative of loss of acinar cell-identity.**
(PDF)

**S5 Table. Quantification of regulated genes for each dTF from Chipseq and RNAseq results.**
(PDF)

**S6 Table. Calculation of acinar dedifferentiation indices for the dtf-cKOs.**
(PDF)

**S7 Table. Comparison of the results of differential expression analysis for previous and new Ptf1a-cKO and Nr5a2-cKO RNAseq datasets.**
(PDF)

**S8 Table. Antibodies for immunofluorescence and chromatin immunoprecipitation.**
(PDF)

## Acknowledgments

The authors thank Jane Johnson for helpful discussions, rabbit anti-mouse Ptf1a antibody for ChIPseq, and a reading of the manuscript, as well as Chris V.E. Wright for the gift of the rabbit anti-mouse Ptf1a antibody.

## Author Contributions

**Conceptualization:** Ana Azevedo-Pouly, Galvin H. Swift, Thomas M. Wilkie, L. Charles Murtaugh, Raymond J. MacDonald.

**Funding acquisition:** Galvin H. Swift, Raymond J. MacDonald.

**Investigation:** Ana Azevedo-Pouly, Michael A. Hale, Galvin H. Swift, Chinh Q. Hoang, Tye G. Deering, Jumin Xue, Raymond J. MacDonald.

**Methodology:** Ana Azevedo-Pouly, Michael A. Hale, Galvin H. Swift, Chinh Q. Hoang, Tye G. Deering.

**Supervision:** Raymond J. MacDonald.

**Writing – original draft:** Galvin H. Swift, Raymond J. MacDonald.

**Writing – review & editing:** Ana Azevedo-Pouly, Michael A. Hale, Thomas M. Wilkie, Raymond J. MacDonald.

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
