## [Decision Letter · Decision Letter 0]

29 Jun 2023

PONE-D-23-11890Key transcriptional effectors of the pancreatic acinar phenotype and oncogenic transformationPLOS ONE

Dear Dr. MacDonald,

Thank you for submitting your manuscript to PLOS ONE. After careful consideration, we feel that it has merit but does not fully meet PLOS ONE’s publication criteria as it currently stands. Therefore, we invite you to submit a revised version of the manuscript that addresses the points raised below during the review process by the reviewers.

We look forward to receiving your revised manuscript.

Kind regards,

Surinder K. Batra

Academic Editor

PLOS ONE

Journal Requirements:

Reviewers' comments:

Reviewer's Responses to Questions

**Comments to the Author**

1. Is the manuscript technically sound, and do the data support the conclusions?

Reviewer #1: Yes

Reviewer #2: Yes

2. Has the statistical analysis been performed appropriately and rigorously? 

Reviewer #1: Yes

Reviewer #2: Yes

3. Have the authors made all data underlying the findings in their manuscript fully available?

Reviewer #1: Yes

Reviewer #2: Yes

4. Is the manuscript presented in an intelligible fashion and written in standard English?

Reviewer #1: Yes

Reviewer #2: Yes

5. Review Comments to the Author

Reviewer #1: The authors of the current manuscript present a descriptive study to describe the role four DNA-binding transcription factors in transcription regulation using a detailed transcriptional approach. While the study does a great job in describing these roles, there are some questions that would still need clarification:

1. While the identification of the four TFs is explained really well in the results, an explanation of the study design at the start of the results section will help the flow of the manuscript tremendously. This can be added as text or as a flow chart in the supplementary figures.

2. Figures 1c and 1g are missing axis labels and need an explanation of what the change values signify- is this a fold change value? Is it an absolute expression value?

3. The transition in between the various sections of the results section is rather abrupt and requires significant changes to help readers follow the study properly. Authors should consider adding explanations to next steps and why they decided to pursue a specific direction. As an example, the switch from study metabolic differences to transcriptomics has no explanation of transition.

While the extensive work the authors have put in is well appreciated, in the current

the manuscript is extremely hard to follow. The authors should consider revising the text extensively in order for it to have the anticipated impact in the field.

Reviewer #2: This manuscript does an excellent job demonstrating the roles of four DNA-binding transcription factors dTFs (Ptf1a, Nr5a2, Foxa2, and Gata4) on the differentiation state of pancreatic acinar cells (PACs). In addition, the study showed the regulatory effects of the dTFs cKO with its effects on KRAS oncogenic transformation (G12D)

Minor edit: There were several abbreviations mentioned in the main text or figure legend without full names (EST, MDS, and CPDB); I would advise the authors to include the full name.

6. PLOS authors have the option to publish the peer review history of their article (what does this mean?). If published, this will include your full peer review and any attached files.

Reviewer #1: No

Reviewer #2: No

---

## [Author Response · Author response to Decision Letter 0]

14 Jul 2023

We thank the reviewers for their expertise and helpful desire to improve the manuscript. 

We have responded to all the reviewers’ comments, below. 

REVIEWER 1

1. “While the identification of the four TFs is explained really well in the results, an explanation of the study design at the start of the results section will help the flow of the manuscript tremendously. This can be added as text or as a flow chart in the supplementary figures.”

As requested by this reviewer, we add a supplemental figure (Figure S1) that outlines the decision tree to identify the four differentiation TFs and the flow/rationale for analyses of the roles they play in defining pancreatic acinar identity and differentiation status. 

2. Figures 1c and 1g are missing axis labels and need an explanation of what the change values signify is this a fold change value?

We have added notations in the legend of Figure 1 that clarify that the axes represent the “# of genes” affected by each dTF cKO. 

3. The transition in between the various sectons of the results secton is rather abrupt and requires significant changes to help readers follow the study. As an example, the switch from study metabolic differences to transcriptomics has no explanation of transition.

We have improved the transition between metabolic studies to transcriptional regulation by inserting two transition sentences at the end of the Metabolism section: 

"To summarize the transcriptomic results: the four dTFs control a very large fraction of the acinar transcriptome by activating the expression of acinar specific genes and by selectively altering the expression of many commonly expressed genes to empower specific cellular processes greatly enhanced in acinar cells. Indeed, the specialized acinar cell functions demand high metabolic activity. To delve into the nature of their collaborative regulatory strategies, we next analyzed the genetic interactions among the dTF genes and examined the role their binding plays at the transcriptional enhancer of a key target gene (Bhlha15/Mist1) as an illustrative example." 

REVIEWER 2

There were several abbreviations mentioned in the main text or figure legend without full

names (EST, MDS, and CPDB); I would advise the authors to include the full name.

We have corrected these oversights. 

In response to the Journal Requirements #2, we have removed the phrase “data not shown”, because these data are not a core part of the research in this study.

---

## [Decision Letter · Decision Letter 1]

31 Aug 2023

Key transcriptional effectors of the pancreatic acinar phenotype and oncogenic transformation

PONE-D-23-11890R1

Dear Dr. MacDonald,

We’re pleased to inform you that your manuscript has been judged scientifically suitable for publication and will be formally accepted for publication once it meets all outstanding technical requirements.

Kind regards,

Surinder K. Batra

Academic Editor

PLOS ONE

Additional Editor Comments (optional):

Reviewers' comments:

Reviewer's Responses to Questions

**Comments to the Author**

1. If the authors have adequately addressed your comments raised in a previous round of review and you feel that this manuscript is now acceptable for publication, you may indicate that here to bypass the “Comments to the Author” section, enter your conflict of interest statement in the “Confidential to Editor” section, and submit your "Accept" recommendation.

Reviewer #1: All comments have been addressed

Reviewer #2: All comments have been addressed

2. Is the manuscript technically sound, and do the data support the conclusions?

Reviewer #1: Yes

Reviewer #2: Yes

3. Has the statistical analysis been performed appropriately and rigorously? 

Reviewer #1: Yes

Reviewer #2: Yes

4. Have the authors made all data underlying the findings in their manuscript fully available?

Reviewer #1: Yes

Reviewer #2: Yes

5. Is the manuscript presented in an intelligible fashion and written in standard English?

Reviewer #1: Yes

Reviewer #2: Yes

6. Review Comments to the Author

Reviewer #1: (No Response)

Reviewer #2: This manuscript does an excellent job demonstrating the roles of four DNA-binding transcription factors dTFs (Ptf1a, Nr5a2, Foxa2, and Gata4) on the differentiation state of pancreatic acinar cells (PACs). In addition, the study showed the regulatory effects of the dTFs cKO with its effects on KRAS oncogenic transformation (G12D). The authors did a great work addressing the comments.

7. PLOS authors have the option to publish the peer review history of their article (what does this mean?). If published, this will include your full peer review and any attached files.

Reviewer #1: No

Reviewer #2: No

---

## [Editor Report · Acceptance letter]

28 Sep 2023

PONE-D-23-11890R1 

Key transcriptional effectors of the pancreatic acinar phenotype and oncogenic transformation 

Dear Dr. MacDonald:

I'm pleased to inform you that your manuscript has been deemed suitable for publication in PLOS ONE. Congratulations! Your manuscript is now with our production department. 

Kind regards, 

on behalf of

Prof. Surinder K. Batra 

Academic Editor

PLOS ONE